# Reward Model Overoptimisation in Iterated RLHF

## Abstract

Reinforcement learning from human feedback (RLHF) is a widely used method for aligning large language models with human preferences. However, RLHF often suffers from reward model overoptimisation, in which models overfit to the reward function, resulting in non-generalisable policies that exploit the idiosyncrasies and peculiarities of the reward function. A common mitigation is *iterated RLHF*, in which reward models are repeatedly retrained with updated human feedback and policies are re-optimised. Despite its increasing adoption, the dynamics of overoptimisation in this setting remain poorly understood. In this work, we present the first comprehensive study of overoptimisation in iterated RLHF. We systematically analyse key design choices: how reward model training data is transferred across iterations, which reward function is used for optimisation, and how policies are initialised. Using the controlled AlpacaFarm benchmark, we observe that overoptimisation tends to decrease over successive iterations, as reward models increasingly approximate ground-truth preferences. However, performance gains diminish over time, and while reinitialising from the base policy is robust, it limits optimisation flexibility. Other initialisation strategies often fail to recover from early overoptimisation. These findings offer actionable insights for building more stable and generalisable RLHF pipelines.

## 1 Introduction

Reinforcement learning from human feedback (RLHF) has become the standard method for aligning large language models with human preferences (Ziegler et al., 2020; Ouyang et al., 2022; Bai et al., 2022). However, RLHF faces a critical vulnerability: reward model overoptimisation (Gao et al., 2023). As fine-tuning progresses, models learn to overfit to the trained reward function - achieving high scores without genuinely satisfying human intent. This creates brittle policies that exploit loopholes rather than developing robust behaviours, leading to systems that appear aligned during training but fail catastrophically when deployed. Iterated RLHF represents a promising approach to combat this problem. By repeatedly collecting new preferences on the latest policy outputs, retraining the reward model, and fine-tuning the policy (Bai et al., 2022; Xiong et al., 2024), practitioners aim to iteratively close the gap between proxy and true reward. Despite its widespread adoption in industry (Ziegler et al., 2020; Ouyang et al., 2022; Bai et al., 2022), it remains uncertain whether iterated RLHF genuinely resolves overoptimisation, merely postpones the inevitable exploitation of the reward model akin to persistent adversarial policies (Gleave et al., 2020), or perpetuates a recurring cycle of overoptimisation in different forms (Singhal et al., 2024).

In this work, we present the first systematic investigation into reward model overoptimisation in iterated RLHF. We identify three pivotal design choices, highlighted in Figure 1, that critically influence the success or failure of the process: *preference data management* (i.e., whether to aggregate or isolate preference data across iterations), *reward function formulation* (i.e., the choice of reward signal to optimize in subsequent training rounds), and *policy initialisation* (i.e., the strategy for initialising the policy at the start of each fine-tuning cycle).

Our key contributions can be summarised as:

- We present the first formal investigation of overoptimisation dynamics across multiple RLHF iterations, relaxing assumptions made in previous work.

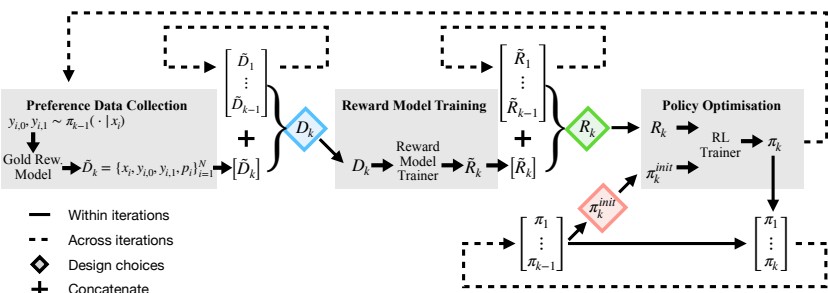

Figure 1: The Iterated RLHF framework performs multiple rounds of preference data collection, reward modelling, and policy optimisation. Our research reveals three design choices that dramatically impact performance: (1) how preference data is managed across iterations, (2) which reward function formulation to optimise, and (3) how policies are initialised at each stage. Effectively configuring these elements can significantly reduce overoptimisation.

- We discuss a systematic evaluation of key design choices with quantitative evidence of their impact on performance and overoptimisation.
- We provide practical guidelines for implementing iterated RLHF, including specific recommendations for preference data management, reward function selection, and policy initialisation strategies.

Using a gold-standard reward model to simulate human labellers (Coste et al., 2024; Gao et al., 2023) on the AlpacaFarm dataset (Taori et al., 2023) and working exclusively with open-source models, our experiments yield several key insights: Reward models become increasingly robust across iterations, leading to higher gold reward scores (Figure 2). Performance gains diminish after three iterations for most methods. Concatenating preference data across iterations dramatically outperforms other approaches. Small but persistent overoptimisation remains after four iterations regardless of design choices.

Our results demonstrate that while iterated RLHF significantly improves reward model robustness, it does not fully eliminate overoptimisation. This underscores the need for continued research into more robust alignment methods that can withstand sophisticated specification gaming (Krakovna et al., 2020) by increasingly capable language models.

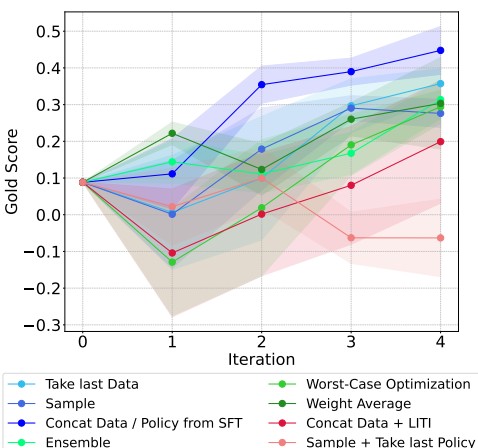

Figure 2: Iterated RLHF design choices in preference data management, reward function formulation, and policy initialization strongly affect ground truth performance and overoptimisation.

## 2 RELATED WORK

RLHF is the standard for aligning large language models to human preference data. The iterated approach has been first discussed by Bai et al. (2022) to fix robustness and calibration issues, attributed to lack of data in the high score regime and has since gained in popularity (Ramé et al., 2024a; Xiong et al., 2024; Ye et al., 2024; Adolphs et al., 2023; Dong et al., 2024; Yuan et al., 2024). Besides training on newly collected preferences, an iterated scheme to train reward models from synthetically generated preference data has been proposed by Wang et al. (2024) and shown to improve performance on the reward model benchmark RewardBench (Lambert et al., 2024), but the authors focus on iterated training of an evaluator and do not study overoptimisation nor the design choices we consider. In the context of Direct Preference Optimisation (DPO) (Rafailov et al., 2023) offline, online and hybrid approaches repeatedly collecting new preference data have been investigated mostly in terms of sample efficiency (Xiong et al., 2024; Das et al., 2024; Muldrew

et al., 2024; Mehta et al., 2023). More broadly iterated methods have been investigated for machine teaching (Liu et al., 2017) and to resolve feedback loops caused by model deployment in supervised learning (Perdomo et al., 2020) and also performative RL (Mandal et al., 2023).

Overoptimisation is a common issue in RL, and evidence of this has been frequently reported in the RLHF literature (Ziegler et al., 2020; Stiennon et al., 2020; Gao et al., 2023; Singhal et al., 2024). A promising method to mitigate overoptimisation is using reward model ensembles combined with conservative optimisation (Coste et al., 2024). Several works further explore reward model ensembles in RLHF (Eisenstein et al., 2024; Lou et al., 2024). Notably, Ramé et al. (2024b) introduce weight averaged reward models (WARM) alleviating the inference cost of multiple reward models during training. Following Coste et al. (2024) and Gao et al. (2023) in tackling reward model overoptimisation, several works propose alternative approaches including reward model distillation (Fisch et al., 2024), hidden state regularisation (Yang et al., 2024b), and more (Yang et al., 2024a; Miao et al., 2024; Liu et al., 2024; Gorbatovski et al., 2024). One commonly reported mode of overoptimisation is length bias (Singhal et al., 2024; Park et al., 2024), which can be tackled by disentangling reward signals related to response length from content quality (Chen et al., 2024). To the best of our knowledge, the literature lacks a systematic investigation into overoptimisation in iterated Reinforcement Learning from Human Feedback (RLHF). Such an investigation is not only necessary but also fundamentally important for a deeper understanding and meaningful improvement of fine-tuning methods based on this technique.

## 3 ITERATED REINFORCEMENT LEARNING FROM HUMAN FEEDBACK

In this section, we first outline the process of a single iteration of RLHF and then extend it to the iterated framework. The RLH pipeline consists of the following three steps: 1. Collection of a preference dataset; 2. Reward model training; 3. Policy optimisation on the reward model. Though not an integral part of the RLHF pipeline, it is common in practice for step 1 to be preceded by supervised fine-tuning on labelled examples. To strengthen our investigation we developed a supporting theoretical framework based on performative prediction (Perdomo et al., 2020) that is presented in Appendix A.

### 3.1 SINGLE-ITERATION RLHF

**Preference data collection.** We start from a supervised fine-tuned policy $\pi^{sft}$ (a policy checkpoint) and use it to collect preference data. The dataset $\mathcal{D}$ contains tuples $\{x_i, y_{i,0}, y_{i,1}, p_i\}$ for $i = 1, ..., N$, where $x_i \in \mathcal{X}$ is a prompt, $y_{i,j} \sim \pi^{sft}(\cdot|x_i)$ for $j = 0, 1$ are two responses from $\pi^{sft}$, and $p_i$ indicates whether $y_{i,0}$ is preferred over $y_{i,1}$. Following Coste et al. (2024) and Gao et al. (2023), preferences $p_i$ are simulated using a gold reward model $R^\star$, which is significantly larger in terms of parameter size than the proxy reward models, serving as approximation for human labels in RLHF. This use of the gold reward model is the standard approach for investigating overoptimisation without incurring significant costs and time bottlenecks due to human labelling. In Appendix E we conduct additional experiments with 25% label noise in the preference data.

**Reward model training.** The proxy reward model $R_\phi$ is initialised from model checkpoint $R^{\text{init}}$, with a randomly initialised prediction head, and subsequently trained by minimizing the cross-entropy loss on the preference dataset $\mathcal{D}$. It is standard to use the Bradley-Terry model (Bradley & Terry, 1952), under which the probability of preferring the answer $y_0$ over $y_1$ given prompt $x$ is given by

$$\mathbb{P}(y_0 \succ y_1|x) = \frac{1}{1 + \exp(R(x, y_1) - R(x, y_0))}. \tag{1}$$

**Policy optimisation.** Having trained the proxy reward model $R_\phi$, the policy $\pi_\theta$ is initialised from $\pi^{sft}$ and then fine-tuned to optimise $R_\phi$. This is commonly achieved with the proximal policy optimization (PPO) algorithm (Schulman et al., 2017). In order to prevent overoptimisation of the proxy reward model and regularise $\pi_\theta$ to not diverge too drastically from its initialisation, a Kullback-Leibler divergence (KL) penalty is used. This yields the overall reward maximised as

$$R^{\text{PPO}}(x, y) = R_\phi(x, y) - \beta \log\left[\frac{\pi_\theta(y \mid x)}{\pi^{sft}(y \mid x)}\right], \tag{2}$$

where $\beta$ controls the strength of the KL penalty (unless specified otherwise we set $\beta = 1 \times 10^{-4}$). This procedure, which only collects preferences once in the entire pipeline, has an important disadvantage. Reward models have been found to be poorly calibrated in the higher reward regime (Bai et al., 2022)

**Algorithm 1** Iterated RLHF (design choices highlighted)

1: **Inputs:** Prompt dataset $X = \{x_i\}_{i=1}^N$, $\pi^{sft}, R^{init}, R^\star$, # of iterations $n_{iter}$
2: $\pi_0 \leftarrow \pi^{sft}$
3: **for** $k = 1$ **to** $n_{iter}$ **do**
4:      $y_{i,0}, y_{i,1} \sim \pi_{k-1}(x_i) \ \forall x_i \in X$
5:      $p_i \leftarrow R^\star(x_i, y_{i,0}, y_{i,1}) \ \forall x_i \in D$
6:      $\tilde{\mathcal{D}}_k \leftarrow \{x_i, y_{i,0}, y_{i,1}, p_i\}_{i=1}^N$
7:      $\mathcal{D}_k \leftarrow \text{CombineData}([\tilde{\mathcal{D}}_1, ..., \tilde{\mathcal{D}}_k])$
8:      $\tilde{R}_k \leftarrow \text{TrainRM}(R^{init}, \mathcal{D}_k)$
9:      $R_k \leftarrow \text{CombineRM}([\tilde{R}_1, ..., \tilde{R}_k])$
10:     $\pi_k^{init} \leftarrow \text{Combine}\Pi([\pi_0, ..., \pi_{k-1}])$
11:     $\pi_k \leftarrow \text{TrainRL}(\pi_k^{\text{init}}, R_k)$
12: **end for**
13: **return** $\pi_k$

Figure 3: Design choices for Iterated RLHF (Algorithm 1). Options include how to combine preference data (latest only, concat, or sample), transfer reward models (last, ensemble, or weight averaged), and initialize policies (last, interpolate, or from SFT). These choices determine how learning signals are propagated through each iteration.

and trained policies overoptimise the proxy reward model leading to unstable fine-tuned policies (Rafailov et al., 2024; Gao et al., 2023; Ziegler et al., 2020). Notably, policy optimization induces a divergence between the distributions $\pi_\theta(y|x)$ and $\pi^{\text{sft}}(y|x)$. This causes the optimised policy to generate outputs that are different from those seen in the training data $\mathcal{D}$. As a result, the reward model $R_\phi$, which was trained on the data $\mathcal{D}$, is now being evaluated on data that it has not seen before.

## 3.2 ITERATED RLHF AND DESIGN CHOICES

The problem of the divergence between the distributions $\pi_\theta(y|x)$ and $\pi^{sft}(y|x)$ is the one addressed by iterated RLHF. In this process, multiple iterations of steps 1-3 of the RLHF pipeline (namely, collection of preference data, reward model training, and policy optimisation)are repeated as shown in Figure 1. Just as in the single-iteration setting, we start from the checkpoint $\pi^{sft}$ and initialise the reward model from $R^{init}$ with a randomly initialised prediction head. However, there are multiple design choices to be made when choosing how exactly to perform iterated RLHF training. We now describe the process in more detail, highlighting the design choices throughout. Please refer to Algorithm 1 for a schematic of the entire process. For simplicity of notation, we omit explicit references to the policy and reward model parameters $\theta$ and $\phi$, using the subscript $k$ to index iterations instead. During the $k^{th}$ iteration of RLHF, we use the policy from the previous one, denoted by $\pi_{k-1}$ to synthesise pairs of responses for the new preference data denoted by $\tilde{\mathcal{D}}_k$.

Indeed, using all policies is unnecessary as it equates to reapplying preference data, but at a higher cost. This new data enables the training of a proxy reward model for which the current policy's output is in-distribution, potentially mitigating the issue of overoptimisation. Taking into account previous iterations, we have access to the list of preference data $[\tilde{\mathcal{D}}_1, ..., \tilde{\mathcal{D}}_k]$. Here we face the first design choice:

*How do we combine the list of $k$ preference datasets into a single training dataset $\mathcal{D}_k$?*

**Combining preference data.** Given a list of $k$ preference datasets, the responses in each of which have been generated by different policies $\pi_1, ..., \pi_{k-1}$, we identify three possible options to consolidate them into a single training dataset. The first option (Figure 3.a) is to simply set $\mathcal{D}_k = \tilde{\mathcal{D}}_k$, only training the reward model on the preference data collected in the current iteration. The second option at the other extreme (i.e., no inter-iteration transfer) is to concatenate all datasets (Figure 3.b). Reusing all the data at each iteration is expected to result in decreased overoptimisation and better approximation with respect to the true reward function. However, this comes with a reward model training computational cost that scales linearly with the number of iterations. Finally, balancing training time and information transfer, we keep the size of the reward model training data constant

across iterations by sampling a subsets $\tilde{\mathcal{D}}_i$ for $i = 1, ..., k$ and concatenating the subsets to form $\mathcal{D}_k$ (Figure 3.c). Once the training data $\mathcal{D}_k$ has been obtained, the proxy reward model $\tilde{R}_k$ can be trained on it. $\tilde{R}_k$ is initialised from the same base model in all iterations. Having trained the reward model, we now arrive at the second critical design choice:

> *How do we transfer information from the list of all previously trained proxy reward models* $[\tilde{R}_1, ..., \tilde{R}_k]$ *into a single reward function $R_k$ that can be optimised by the policy?*

**Combining reward models.** The reward model is the crucial piece in obtaining generalisable and consistent policies in RLHF, and it is even more important over multiple iterations as effects compound. Given the list $[\tilde{R}_1, ..., \tilde{R}_k]$ containing the $k$ proxy reward models leading up to the current iteration the task is to obtain a robust reward function to be optimised. We note that this design choice can be considered in parallel to the combination of preference data, as both target the same outcome of transferring information from previous iterations to the reward function.

To achieve this task we investigate three types of solutions. The first only uses the most recently trained proxy reward model setting $R_k = \tilde{R}_k$ (Figure 3.d), hence there is no utilisation of previously trained reward models. In contrast, the second option ensembles all previously trained proxy RMs taking the mean of the individual rewards (Figure 3.e) (Coste et al., 2024). Since reward model ensembles showed limited improvements in Coste et al. (2024) we also evaluate worst-case optimisation (WCO), i.e., optimising the minimum $R_k(x, y) = \min_{i=1,...,k} \tilde{R}_i(x, y)$. This option comes with the disadvantage of requiring inference on $k$ reward models in parallel. To address the computational cost, we also consider weight averaged reward models (see Figure 3.f) by performing task arithmetic (Ilharco et al., 2023). More formally, given a sequence of reward models $\tilde{R}_1, ..., \tilde{R}_k$, which are parameterised by $\tilde{\phi}_1, ..., \tilde{\phi}_k$, respectively, we obtain the proxy reward function $R_k$ parameterised by $\phi_k$ as follows: The ensemble uses $R_k(x, y) = \frac{\sum_{i=1}^{k} \tilde{R}_i(x,y)}{k}$ and to obtain the weight averaged reward model we set $\phi_k = \frac{\sum_{i=1}^{k} \tilde{\phi}_i}{k}$. Having obtained the reward function, the next and final step of each iteration is to optimise it, which leads us to the third and final design choice:

> *Given $\pi^{sft}$ and the fine-tuned policies $\pi_1, ..., \pi_{k-1}$, how can we choose $\pi_k^{init}$ to balance efficiency and robustness against overoptimisation?*

**Policy initialisation.** The final design choice concerns the initialisation of the policy, i.e., how $\pi_k^{init}$ is chosen. Bai et al. (2022) initialise the policy from $\pi^{sft}$ at every iteration, not taking into consideration previously performed computation. We call this initialisation *From SFT* shown in Figure 3.i. As alternative, we use linear interpolation towards initialisation (*LITI*) (Ramé et al., 2024a), which was inspired by WiSE-FT proposed by (Wortsman et al., 2022). With *LITI*, shown in Figure3.h, we set $\pi_k^{init} = (1 - \eta)\pi_{k-1}^{init} + \eta\pi_{k-1}$, where $\eta$ is a hyperparameter balances the optimisation of $R_{k-1}$. Taking $\eta = 1$ corresponds to initialising the current policy from the previously fine-tuned one, setting $\pi_k^{init} = \pi_{k-1}$. Since continuing fine-tuning of the most recent policy fully relies on the previous iterations, it may suffer from entropy collapse leading to no optimisation in later iterations. Continuing with the fine-tuned policy carries risks if undesirable behaviour learned in previous iterations cannot be unlearned. Note, when performing *LITI*, the policy is regularised with the KL between the policy and its initialisation $\pi_k^{init}$.

## 4 EVALUATING OVEROPTIMISATION IN ITERATED RLHF

In Section 3 we formalised the process of iterated RLHF and highlighted the critical design choices. In this section, we detail our evaluation setup, emphasizing the quantification of overoptimisation and examining how its progression over iterations is influenced by different design choices.

**Training setup.** Our evaluation setup follows extensive prior works that study overoptimisation in the single iteration RLHF in a controlled and simulated manner (Coste et al., 2024; Gao et al., 2023). Similarly to Coste et al. (2024) we use instructions from the *AlpacaFarm* dataset (Dubois et al., 2023) for reward model training and policy optimisation. The preference data $\tilde{\mathcal{D}}_k$ collected at each iteration contains preferences for a subset of 1000 instructions in the preference split of AlpacaFarm. Preference labels $p_i$ are simulated with the 7 billion parameter `Reward-Model-AlpacaFarm-Human` (Dubois et al., 2023), which is also used by Coste et al. (2024). It is worth noting again the significant difference in parameter sizes between the proxy reward models and the gold reward

model, justifying the use of the gold reward model as a proxy for human labellers. Similarly to Coste et al. (2024), to obtain $\pi^{sft}$, we performed supervised fine-tuning on the `pythia-410m` model (Biderman et al., 2023) on the AlpacaFarm SFT split. We chose `pythia-410m` as it achieves an appropriate balance between computational cost and experimental rigour for our investigation. Gao et al. (2023) also found that policy size did not affect the shape of the overoptimisation curve in their setting, further justifying this choice of policy. We initialise proxy reward models $\tilde{R}_k$ from the HuggingFace checkpoint `pythia_70m_sft` provided by Coste et al. (2024), as well as the larger `pythia-160m`, with a randomly initialised prediction head (Coste et al., 2024). We train reward models for 5 epochs with a learning rate of $1 \times 10^{-5}$ (Coste et al., 2024). For policy optimisation, we perform 6000 steps of PPO on the unlabelled split of AlpacaFarm. The learning rate is set to $1 \times 10^{-6}$ and a constant KL penalty of $1 \times 10^{-4}$ is used. The full specifications of the hyperparameters for reward model training and policy optimisation, and the prompt format are given in Appendix C.

We perform a total of 4 iterations per method and report the results of the final iteration in comparison to the initial one. All results presented in our performance evaluation are reported for 8 random seeds, except for policy initialisation *From SFT* with the *Take last* configuration for both preference data and reward model, for which we only obtained 4 random seeds due to compute constraints. We note that this is still above the commonly reported 3 random seeds. To aggregate seeds in both gold score and KL we collect all seeds per iteration, bucket data points by KL. We then plot the mean and standard deviation of the gold rewards per bucket against the KL.

**Measuring overoptimisation with the Maximum Mean Discrepancy.** The standard methodology for investigating reward model overoptimisation is to compare mean rewards on proxy vs. gold reward functions over a hold-out set (Coste et al., 2024; Moskovitz et al., 2024; Gao et al., 2023). This overlooks discrepancies in the high-reward tail, which more strongly influence policy optimisation. We instead compare reward models by their distributions of rewards, evaluating on the 2000 unseen instructions contained in the validation split of AlpacaFarm at every 300 steps during policy optimisation.

Our approach to measuring differences between reward functions consists of two steps, the first of which is a standardisation that ensures reward functions that lead to the same ordering of policies when optimised are treated as equal (see Appendix B.1). In the second step, we use the maximum mean discrepancy (MMD) (Gretton et al., 2012) to measure the discrepancy between the two reward functions. In particular, we utilise this method to compare the proxy reward models trained at each iteration with the gold-reward model $R^\star$. For full details and a justification of the validity of this method we refer the reader to Appendix B.

## 5 EXPERIMENTAL RESULTS

When comparing different methods, we primarily focus on their performance in the final iteration, as this iteration consistently outperforms previous ones for all algorithms. Additionally, it demonstrates the reward-KL curves produced by each method. We also compare the performance of methods across multiple iterations, to see how the KL-reward curves change through the iterations.

### 5.1 ITERATED RLHF CAN CLOSE THE GAP BETWEEN PROXY AND GOLD REWARD FUNCTIONS

Before investigating the differences between the design choices, we focus on the progression of reward model robustness across iterations more generally. In Figure 4, we show how performing multiple iterations of RLHF, concatenating all preference data to train the reward model, and re-initialising the policy from $\pi^{sft}$ at each iteration decreases the gap between the gold reward function and the proxy. As iterations progress, the proxy reward model becomes more robust and increasingly aligned with the gold reward model on the distribution observed during policy optimisation.

Furthermore, the KL-reward Pareto front advances with each iteration, although improvements plateau as the distance between proxy and gold reward curves shrinks in later iterations. These performance plateaus appear to result from a combination of interacting factors rather than simple diminishing returns. First, the proxy reward model progressively converges toward the gold reward model on the distribution induced by policy optimisation, which limits the scope for further improvement. Second, policy entropy tends to decline across iterations, particularly when initialisation methods other than *From SFT* are used. Third, data saturation may occur once additional preference data provides little novel information. However, there remains scope to better align gold and proxy reward functions. Comparing reward distributions across iterations further reveals that, after the

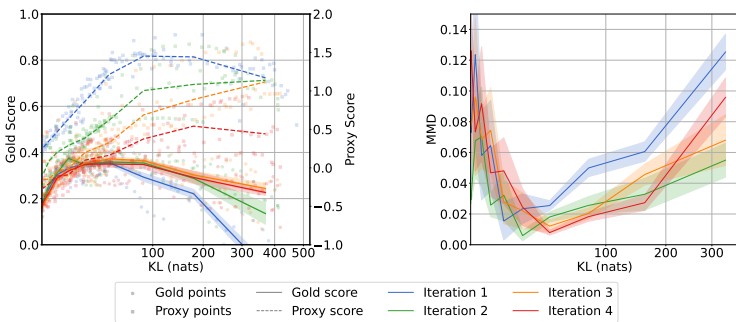

Figure 4: Progression of proxy–gold alignment across RLHF iterations with $\pi^{\text{sft}}$ reinitialisation and concatenated data. Mean scores show narrowing gaps and improved robustness, though with diminishing returns. MMD reveals early convergence but rising divergence at higher KL, highlighting distributional shifts not observed in mean scores.

policy closely approximates the output distribution in $\mathcal{D}_k$, the MMD increases again in the high-KL regime for all iterations, especially rapidly in the initial iteration (see Figure 4). We hypothesize that the non-monotonic relationship between MMD and KL reflects a dynamic interplay between alignment and exploitation during training. Early on, RL against the proxy RM improves alignment with held-out samples from initialisation, reducing MMD as the proxy's predictions grow closer to the gold RM. Later, as the policy distribution shifts and begins exploiting proxy-specific quirks (increasing KL), outputs diverge from true human preferences, driving MMD back up. Additionally, the rate at which the proxy-gold reward gap closes varies considerably among methods (see Appendix F.1), highlighting the importance of investigating design choices described in Section 3.

## 5.2 Combining preference data

**Scaling reward model training data is most effective.** We first focus on comparing methods for combining preference datasets. To isolate the effects of varying the combination strategy, we fix the policy initialization to *From SFT* and reward models are combined using the *Take last* approach. As shown in Figure 5a, all methods demonstrate significant improvements over a single iteration, particularly in preventing performance collapse at higher KL divergences.

The *Concatenate* strategy achieves consistently higher gold scores, especially in the KL range of 50-200 nats. While *Take last* and *Sample* approaches show similar trends and substantial improvements over iteration 1, they do not quite match the performance of full data concatenation. This result is coherent with the finding that increasing training dataset size reduces reward model overoptimisation (Gao et al., 2023), explaining why the sampling strategy is outperformed by concatenating all datasets. A critical observation is that beyond KL $\approx 200$, the baseline iteration 1 experiences severe performance degradation due to overoptimisation, dropping to negative gold scores. In contrast, all iterative approaches maintain positive performance even at high KL values, demonstrating their effectiveness in mitigating overoptimisation. This ranking of methods is not only observed in the final iteration, but is already exhibited as early as the second iteration as shown in Figure 2 and in Appendix F.2.

**Ensuring full coverage of the prompts when sampling matters less.** While the sampling strategy slightly outperformed taking only the newest preference dataset, it did not achieve the same level of performance as concatenating all data. Here we take a closer look at the sampling strategy. In Figure 5b standard sampling with potential prompt repetition (*Sample*) and sampling where each prompt appears exactly once (*Sample Exclusive*). The differences are minor, suggesting that prompt repetition has a limited impact on performance or overoptimisation. This pattern also holds in earlier iterations (Appendix F.2), highlighting that while data combination strategies are effective at preventing overoptimisation, the computational cost of maintaining and training on growing datasets remains, as more efficient methods are unable to achieve the same performance as *Concatenate*. This motivates exploring reward-model combination in parameter space to achieve similar gains with less overhead.

## 5.3 Combining reward models

**No free lunch by merging reward models.** Concatenating all preference data, previously the most effective method, serves as our performance baseline. As shown in Figure 5c, all approaches

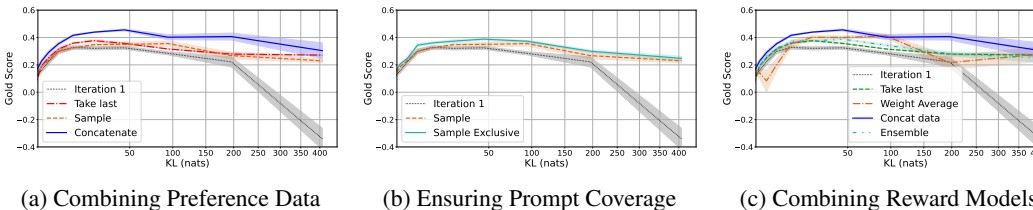

(a) Combining Preference Data      (b) Ensuring Prompt Coverage      (c) Combining Reward Models

Figure 5: Iterated RLHF benefits most from scaling reward model training data. (a) Concatenating all preference data across iterations best mitigates overoptimisation, especially at mid KL (50–200). (b) Sampling, with or without prompt repetition, performs similarly, implying limited impact of prompt coverage. (c) Parameter-space methods (ensembles, averaging) lead to efficiency gains but fall short of the simpler *Take last* with data aggregation.

improve similarly in early KL regions (up to ≈ 50 nats), reaching comparable performance. *Weight Average* and *Ensemble* maintain strong, efficient performance, though ensembles increase inference time and memory use. The mean objective offers no clear gains over the *Take Last* approach with a single reward model, consistent with Coste et al. (2024). More specifically, Figure 5c shows that *Ensembling* does not outperform at the 70M scale. Although we do not explicitly measure calibration, MMD serves as a proxy for calibration in the high-reward tail, suggesting that either the calibration benefits are limited at this scale, or the optimisation with PPO exploits it regardless of calibration. While weight averaging has been reported to outperform ensembles (Ramé et al., 2024b), we only observe differences in the mid-KL regime. In contrast to prior work (Coste et al., 2024; Ramé et al., 2024b), we combine models trained on data with significantly different joint distribution over pairs $(x, y)$. Regardless, both methods still provide significant improvements when comparing the fourth and first iterations. The various reward model combination methods in RLHF perform similarly, suggesting computational efficiency should drive selection.

**Larger reward models benefit more from combining reward models.** We now investigate how scaling the reward model size affects performance in iterative RLHF. While concatenating all preference data with policy initialisation from the SFT checkpoint remains the most robust approach, we observe that alternative reward model strategies benefit significantly from increased reward model capacity. As shown in Figure 6, performance differences between the 70M and 160M reward models are most pronounced for *Ensemble* and *Worst-Case Optimisation*, with both methods substantially improving at the larger scale and approaching the performance of the data concatenation baseline by the fourth iteration. This suggests that while reward model combination methods did not match the effectiveness of preference data concatenation at smaller scales, their potential is unlocked with more expressive reward models. These results highlight that design choices affecting reward model size not only influence individual model accuracy but can significantly enhance the utility of design choices combining reward models in iterated RLHF settings.

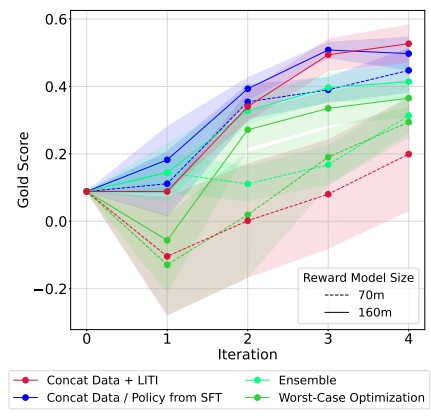

Figure 6: Impact of reward model size on gold score. Larger models (160M, solid) outperform smaller ones (70M, dashed), with the biggest gains in *Ensemble* and *Worst-Case Optimisation*. *From SFT* stays stable, while *LITI* steadily improves with scale.

### 5.4 Policy Initialisation

**Initialising from SFT is the most robust.** Finally, comparing the policy initialisation methods we observe that no method improves on the KL-reward Pareto front achieved by concatenating all preference data and initialising the policy from the SFT checkpoint (Figure 7a). Sampling the preference data is similarly robust, highlighting that initialising with *From SFT* results in generally reduced overoptimisation. Note, *LITI* and *Take last* start from significantly larger KL due the compounding of KL through repeated initialisation increasingly further away from $\pi^{sft}$ in the KL space. Resetting the policy at each iteration combined with the aggregation of preference data results

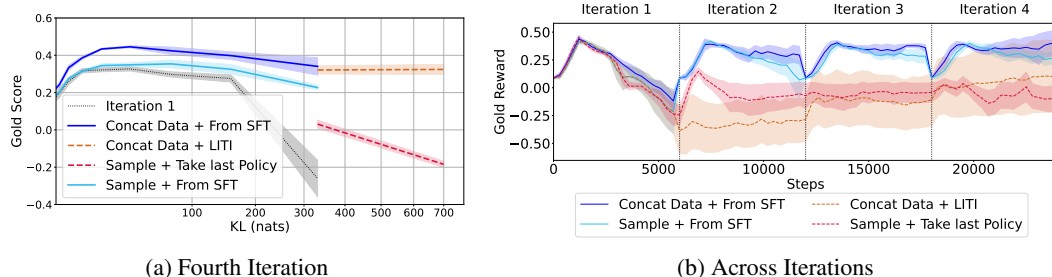

(a) Fourth Iteration            (b) Across Iterations

Figure 7: Effect of policy initialisation on overoptimisation and recovery across iterations. *From SFT* is most robust, avoiding divergence via resets and data aggregation. *LITI* and *Take last* start from high KL due to cumulative divergence. Overoptimised policies are hard to recover: *Take last* worsens over time, while *LITI* improves but does not reach *From SFT*.

in consistently less overoptimisation and more performant policies. Although, initialisation with $\pi^{sft}$ limits the flexibility and potential gains that could be realised by continued optimisation.

**Overoptimised policies are hard to recover from.** While *From SFT* is reset at the end of each iteration, *LITI* and *Take last* have to recover form the initial overoptimisation, as shown in Figure 7b. The behaviour in earlier iterations reveals the consistent performance improvements attained with *LITI*. On the other hand, *Take last* is unable to recover after overoptimising again in the second iteration, despite the counterpart, sampling preference data but initialising *From SFT*, improving with each iteration. This is partly due to decreasing entropy caused by to the prolonged optimisation when using the *Take last* initialisation (see Appendix F.5), the mean gold reward does not exceed zero in the third and fourth iterations. However, the primary failure mode is the policy exploiting weaknesses and idiosyncrasies of the proxy reward models that cannot be corrected in following iterations. In Appendix G we show an example of this behaviour, in which the response consists of narrow, repeated token sequences. Despite *LITI* improving on average across multiple seeds, we observe that linear interpolation is also unable to recover strongly overoptimised seeds (see Appendix F.4). Thus, while *From SFT* is most robust, it is also limited by the repeated initialisation from $\pi^{sft}$.

**Policy interpolation works better with larger reward models.** We hypothesise that *LITI* could achieve similar or higher gold scores than *From SFT* after more iterations. Supporting this, our experiments with a larger reward model show that *LITI* benefits substantially from increased reward model capacity (see Figure 6). This improvement likely stems both from better-calibrated gradients that support recovery, and from the fact that larger reward models tend to overoptimise less aggressively (Gao et al., 2023), resulting in safer intermediate policies and more stable interpolation paths. These findings highlight the importance of early stopping and reward model design when using policy initialisation methods other than *From SFT*, and suggest that *LITI* may become increasingly competitive as reward model expressiveness scales. Despite promising scaling results of *LITI*, *From SFT* initialisation remains the safe option for less expressive reward models.

### 5.5 REWARD MODEL EVALUATION ON REWARDBENCH

We evaluate the reward models from the first and final iterations obtained via the different design choices on RewardBench (Lambert et al., 2024). In particular, since we train on AlpacaFarm, we report the performance on the AlpacaEval Easy and Hard splits as well as the overall accuracy averaged across all subsets of the benchmark.

In Figure 8 we observe that the proxy reward models, which correspond to methods that ultimately achieve high gold reward in the final iteration, obtain higher or comparable accuracy on the AlpacaEval subsets when comparing the first and final iterations. For the larger reward models, this is the case for *Concat Data* as well as for *LITI*. The remaining design choices yield final reward models that achieve lower accuracy on AlpacaEval subsets than their counterparts trained in the initial iteration. The full results, also including the 70M reward models, can be found in Appendix D.

While this evaluation on RewardBench provides an additional perspective on the generalization of the proxy reward models, we note that RewardBench performance is less diagnostic for our setting than gold-reward alignment. Due to distribution shift, reward models in later iterations may effectively mitigate overoptimisation even if their accuracy on the RewardBench test set is lower.

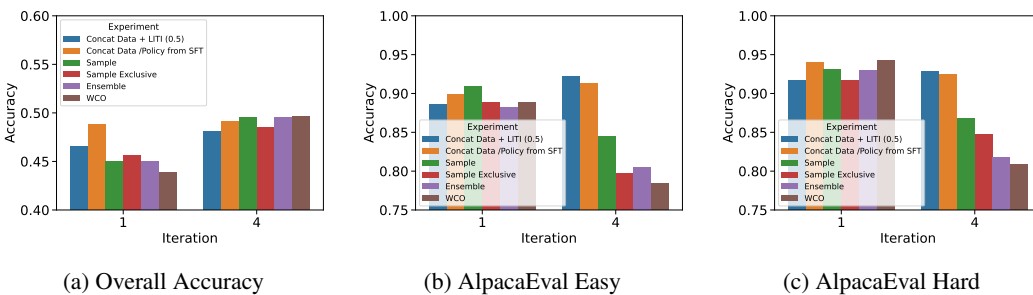

Figure 8: RewardBench performance for 160M reward models.

## 6 LIMITATIONS

Our study focuses on controlled settings, using modestly sized policy models (Pythia-410M) and reward models (70M, 160M) on the AlpacaFarm benchmark with a static "gold" reward model to simulate human feedback. However, this setup, consistent with prior work (Coste et al., 2024; Ramé et al., 2024b; Zhu et al., 2024) (also in terms of model size), enables systematic investigation of iterative RLHF while ensuring results remain interpretable and comparable. Moreover, scaling laws suggest policy size is not the main driver of overoptimisation and that scale effects are smooth (Gao et al., 2023), indicating that our findings and design choices are likely to transfer to larger models, even if the degree or speed of overoptimisation differs. Using a single dataset (AlpacaFarm) enabled controlled ablations but may not capture the diversity of real-world tasks. We also note that a static "gold" reward model, abstracts away the noisy and evolving nature of human preferences. However, this is standard practice in the field (Coste et al., 2024; Gao et al., 2023) to ensure reproducibility and mitigate cost of human labelling. In Appendix E, we conduct experiments with simulated label noise, which show that our conclusions extend to this more realistic setting. Preference drift remains an interesting problem that is beyond the scope of this paper. We ran four iterations, enough to observe plateaus and overoptimisation trends, but further scaling was prohibitive given the computational resources of our institution. Nonetheless, our work offers key insights and guidance for designing more robust iterative RLHF pipelines and lays groundwork for future research on larger scales and real-world settings.

## 7 CONCLUSION

In this work we have presented the first investigation of reward model overoptimisation in iterated RLHF. Through simulations with a gold-standard reward model and analysing distributional discrepancies, we have demonstrated that overoptimisation diminishes across iterations as reward models better approximate the ground truth. However, improvements begin to plateau after three iterations. Consequently, we recommend prioritizing a smaller number of iterations with *Concat Data* rather than attempting to match performance by extending cheaper strategies such as *Take Last*, as the latter incurs a considerably higher overall cost. It is worth noting that larger reward models exhibit stronger performance when using *Ensemble* and *WCO* strategies. While completely eliminating overoptimisation remains unattainable, we have identified base policy initialisation as the most robust approach, despite its reduced optimisation flexibility. Our analysis provides practical guidelines for implementing iterated RLHF and illuminates fundamental trade-offs in iterative preference learning, establishing a foundation for future research in reliable RLHF systems.

## ETHICS STATEMENT

We have carefully considered the broader impact of this work. Our research focuses on overoptimisation in reinforcement learning from human feedback (RLHF), which is an important area for improving alignment between AI systems and human preferences. While RLHF as a field has potential implications for fairness, bias, and the societal impact of large-scale deployment, the contributions in this paper are methodological and do not involve sensitive data, human subjects, or direct deployment in real-world applications. We therefore do not anticipate any immediate ethical concerns arising directly from this work.

## REPRODUCIBILITY STATEMENT

We have taken several steps to ensure the reproducibility of our work. All models and benchmarks used in our study are open source, with appropriate links and licensing information provided. Detailed descriptions of the training procedures are presented in Section 4, and the full set of hyperparameters as well as the prompts used during training are reported in Appendix C. To account for variability, all experiments are conducted across eight random seeds. We commit to releasing our code upon acceptance of the paper to further facilitate reproducibility.

## LLM USAGE STATEMENT

We used LLMs, in particular ChatGPT and Claude to aide the writing process. Specifically, for paraphrasing and shortening existing paragraphs of the manuscript, as well as polishing the wording of certain paragraphs for clarity.

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

# A A THEORETICAL FRAMEWORK: ITERATED RLHF AND PERFORMATIVE PREDICTION

## A.1 OVERVIEW

We note that the framework of performative prediction (Perdomo et al., 2020) can be applied to our setting. In fact, when performing iterated RLHF, we are simulating performative prediction or more specifically a version of strategic classification. We have that a reward model $R_\phi$ induces a potentially different distribution $\mathcal{D}(\phi)$ over instances $(x, y)$ where continuations $y$ are obtained from the policy $\pi_\theta$ optimised for $R_\phi$, which yields that a reward model $R_{\phi_{PO}}$ is performatively optimal if $\phi_{PO} = \arg\min_\phi \mathbb{E}_{(x,y)\sim\mathcal{D}(\phi)} \ell((x, y, \phi))$. Furthermore, a model $R_{\phi_{PS}}$ is defined as performatively stable if $\phi_{PS} = \arg\min_\phi \mathbb{E}_{(x,y)\sim\mathcal{D}(\phi_{PS})} \ell((x, y, \phi))$. Intuitively, retraining a performatively stable reward model after optimising against it will yield the same reward model. As such the reward model would not be over-optimised and still perform optimally on its induced distribution. In Theorem 3.5 Perdomo et al. (2020) provide 3 conditions under which the reward model obtained from repeated iterations of RLHF converges to a unique performatively stable reward model at a linear rate. We require the loss to be $\beta$-jointly smooth and $\gamma$-strongly convex, and the map $\mathcal{D}(\cdot)$ from reward model parameters to the distribution of prompt continuation pairs to be $\epsilon$-sensitive. Since as part of the map $\mathcal{D}(\cdot)$ the policy is optimised with PPO, where small changes in the reward model can lead to significant changes in the optimal policy, this mapping is generally not $\epsilon$-sensitive. As a consequence, linear convergence is not guaranteed. Note, that we may still aim for close to linear convergence by making adjustments to satisfy the stated conditions.

In the following subsections we expand on the overview above and present a concise theoretical account of iterated RLHF. We use $\phi$ to denote reward-model parameters and $\theta$ to denote policy parameters. Our presentation casts iterated RLHF as a performative prediction problem, and then derives sufficient conditions for convergence as well as a set of practical propositions that explain common empirical mitigations (data aggregation, reward model ensembles, and policy resetting).

## A.2 SETUP

Let $\pi_\theta$ be a stochastic policy parameterised by $\theta \in \Theta$. Let $R_\phi$ be a learned reward model parameterised by $\phi \in \Phi$. We denote by $\Pi(R_\phi) \mapsto \pi_{\theta(\phi)}$ the policy optimisation operator that (approximately) returns a policy optimised with respect to $R_\phi$. Running $\pi_{\theta(\phi)}$ in the environment (or simulator) induces a distribution over prompts and model responses; we write $D(\phi)$ for the resulting distribution over observed preference pairs or *(input, response)* tuples $(x, y)$.

A reward model is trained by empirical risk minimization on data sampled from the distribution induced by the current reward model through policy optimisation. Concretely, given a loss function $\ell(\phi; (x, y))$ (for example cross-entropy or a surrogate for pairwise preference loss), the standard iterated update considered throughout this work can be defined as follows:

$$\phi_{t+1} = \arg\min_\phi \ \mathbb{E}_{(x,y)\sim D(\phi_t)}\big[\ell(\phi; (x, y))\big]. \tag{3}$$

This framing matches the performative prediction viewpoint: the object being learned (the reward model) affects the data distribution through the downstream policy it induces.

## A.3 PERFORMATIVE STABILITY

**Definition A.1** (Performative stability). A reward model parameter $\phi^*$ is called performatively stable if it is a fixed point of the update equation 3, i.e.

$$\phi^* = \arg\min_\phi \ \mathbb{E}_{(x,y)\sim D(\phi^*)}\big[\ell(\phi; (x, y))\big].$$

At a performatively stable point, retraining the reward model on data produced by the policy it induces produces no change. Iterated RLHF can therefore be interpreted as an algorithmic attempt to reach such a fixed point.

## A.4 CONVERGENCE GUARANTEES

**Theorem A.2** (Convergence to a performatively stable point). *Suppose the per-example loss $\ell(\phi; (x, y))$ is $\alpha$-strongly convex and $\beta$-smooth in $\phi$ for every data point $(x, y)$. Suppose further that the mapping $\phi \mapsto D(\phi)$ is $L$-Lipschitz in total variation distance. If $\frac{L\beta}{\alpha} < 1$, then the map defined by the update equation 3 is a contraction and the iterates $\{\phi_t\}_{t \geq 0}$ converge linearly to a unique performatively stable point $\phi^*$.*

*Proof sketch.* For each fixed data distribution, strong convexity and smoothness imply the population risk admits a unique minimizer, and the $\arg\min$ mapping is Lipschitz with constant at most $\beta/\alpha$. Composing this with the $L$-Lipschitz dependence of $D(\phi)$ on $\phi$ yields an overall contraction constant bounded by $L\beta/\alpha$. If this constant is strictly less than one, Banach's fixed point theorem guarantees a unique fixed point and geometric convergence of iterates. This is an application of the performative prediction contraction framework (Perdomo et al., 2020).

**Discussion.** The theorem isolates two failure modes in practice: (i) the loss used to train reward models is rarely globally strongly convex in modern neural parameterisations, and (ii) modern policy optimisers (PPO, SAC, etc.) can induce highly non-Lipschitz changes in the data distribution, i.e., small changes to $\phi$ may yield large shifts in $\pi_{\theta(\phi)}$ and hence in $D(\phi)$. Consequently, the sufficient conditions above are not satisfied in general RLHF pipelines, but they nevertheless clarify why certain regularisers and protections (e.g., constraining policy updates, aggregating data) promote stable behaviour.

## A.5 PREFERENCE DATA AGGREGATION

**Proposition A.3** (Data aggregation reduces estimation error). *Let the reward model be trained by empirical risk minimization on a dataset $\mathcal{S}$ of size $N$. Under standard i.i.d. concentration bounds, the expected generalization error of the empirical minimizer scales as $O(1/\sqrt{N})$. If datasets collected across iterations $\mathcal{S}_1, \ldots, \mathcal{S}_T$ are concatenated to form $\mathcal{S}_{\text{tot}}$ with total size $N_{\text{tot}} = \sum_{t=1}^{T} N_t$, the estimation error correspondingly decreases as $O(1/\sqrt{N_{\text{tot}}})$.*

*Proof sketch.* This follows from Hoeffding-type concentration or uniform convergence arguments: more samples tighten empirical estimates of the population risk and hence reduce the gap between empirical and population minima.

**Corollary A.4.** *Training on aggregated data approximates training on the mixture distribution $D_{\text{mix}} = \frac{1}{T} \sum_{t=1}^{T} D(\phi_t)$, reducing variance and decreasing sensitivity to idiosyncrasies of any single iteration.*

**Discussion.** Aggregation stabilizes training in two ways: it increases effective sample size (reducing estimation noise) and smooths the effective data generating process, which can reduce the Lipschitz constant of $\phi \mapsto D(\phi)$ empirically.

## A.6 REWARD-MODEL ENSEMBLES AND TRANSFER

**Proposition A.5** (Averaging reduces squared error). *Let $R_{\phi_i} = R^* + \delta_i$ be $K$ proxy reward models with additive errors $\delta_i$. Let define the ensemble reward as: $R_{\text{ens}} = \frac{1}{K} \sum_{i=1}^{K} R_{\phi_i} = R^* + \frac{1}{K} \sum_{i=1}^{K} \delta_i$. Then*

$$|R_{\text{ens}} - R^*|_2^2 = \Big| \frac{1}{K} \sum_{i=1}^{K} \delta_i \Big|_2^2 \leq \frac{1}{K} \sum_{i=1}^{K} |\delta_i|_2^2.$$

*Proof sketch.* This is a direct consequence of Jensen's inequality / the variance reduction property of averaging.

**Discussion.** When errors $\delta_i$ are approximately zero-mean and weakly correlated, ensembles can substantially reduce the magnitude of systematic errors that policies can exploit. Worst-case ensemble strategies (e.g., conservative lower-bound ensembles) further limit reward overestimation.

## A.7 POLICY INITIALIZATION AND RESET STRATEGIES

Let $\pi_{\theta_0}$ denote a base supervised fine-tuned (SFT) policy. Let us define the Kullback–Leibler divergence between two policies by $D_{\mathrm{KL}}(\pi_\theta \| \pi_{\theta_0})$.

**Proposition A.6** (Resetting bounds policy drift). *If at every iteration the policy optimisation is initialized from the base policy $\pi_{\theta_0}$ (i.e. we re-start optimisation from $\theta_0$), then the accumulated divergence from the base policy over iterations is bounded by the per-iteration optimisation step sizes. In contrast, warm-starting from the previous iterate $\theta_{t-1}$ can lead to cumulative drift: divergences can add across iterations and become large.*

**Discussion.** Resetting is an effective empirical safeguard against runaway behaviour and can improve reproducibility at the cost of reduced per-iteration adaptivity.

## A.8 OVEROPTIMISATION (ERROR-TO-GAP) BOUND

**Proposition A.7** (Error–to–gap bound). *Suppose the reward model approximation error is uniformly bounded: for all outputs $y$, $|R_\phi(y) - R^*(y)| \le \varepsilon$.*

*Then the suboptimality gap in the maximized rewards satisfies*

$$\max_y R^*(y) - \varepsilon \le \max_y R_\phi(y) \le \max_y R^*(y) + \varepsilon.$$

**Discussion.** Bounding the sup-norm error of the reward model controls the extent to which an optimiser can overestimate the true reward. The preceding propositions (aggregation and ensembling) are practical mechanisms for reducing $\varepsilon$ and hence for limiting overoptimisation.

## A.9 CONCLUDING REMARKS

Framing iterated RLHF as a performative prediction problem clarifies both desirable algorithmic choices and structural failure modes. Under favourable convexity and Lipschitz conditions one recovers a contraction argument guaranteeing convergence to a unique performatively stable reward model. In realistic RLHF pipelines these conditions fail, but the theory explains why mitigation strategies—data aggregation, reward model ensembles, and policy resets—improve stability: they reduce estimation variance, shrink reward-model error, and bound policy drift. Together these tools help iterated RLHF approximate performatively stable equilibria in practice, even when exact theoretical conditions are not met.

## B  REWARD MODEL COMPARISON WITH THE MAXIMUM MEAN DISCREPANCY

Formally, our goal is to compare any two reward functions $R_{\phi_1}$ and $R_{\phi_2}$. As the first step, we scale both reward functions to have mean zero and variance one. This ensures that reward functions, which differ only by an affine transformation, are treated as equal to one another after scaling. For details about this result, please refer to Appendix B.1. This is desirable since affine transformations do not affect the ordering over policies induced by the original and transformed reward functions when they are optimised Skalse et al. (2024).

As the second step, we compute the discrepancy between $R_{\phi_1}$ and $R_{\phi_2}$. While we have reward functions in principle, during training, only samples of rewards from the true and proxy are observed. Given that prompts are identically and independently distributed $x_i \overset{i.i.d.}{\sim} \rho$ and $y_i \sim \pi_\theta(\cdot|x_i)$, we obtain that the observed rewards $r_i = R_\phi(x_i, y_i)$ are i.i.d samples (details in Appendix B.1). As a consequence, we can rely on the Maximum Mean Discrepancy (MMD) to measure the discrepancy between distributions of observed rewards from $R_{\phi_1}$ and $R_{\phi_2}$. The MMD compares two distributions based on their distances in the feature space determined by the chosen kernel. It is known for its strong theoretical guarantees, and it is commonly used in the two sample testing literature (Gretton et al., 2012). We use the popular squared exponential kernel.

Given samples $\mathbf{r}_{\phi_1} := \{r_{\phi_1,1}, ..., r_{\phi_1,n}\}$ and $\mathbf{r}_{\phi_2} := \{r_{\phi_2,1}, ..., r_{\phi_1,n}\}$ an unbiased empirical estimate of the MMD is obtained by

$$
\begin{aligned}
\text{MMD}_u^2[\mathbf{r}_{\phi_1}, \mathbf{r}_{\phi_2}] = & \frac{1}{n(n-1)} \sum_{i=1}^n \sum_{j \neq i}^n k\left(r_{\phi_1,i}, r_{\phi_1,j}\right) \\
& + \frac{1}{n(n-1)} \sum_{i=1}^n \sum_{j \neq i}^n k\left(r_{\phi_2,i}, r_{\phi_2,j}\right) \\
& - \frac{2}{n^2} \sum_{i=1}^n \sum_{j=1}^n k\left(r_{\phi_1,i}, r_{\phi_2,j}\right).
\end{aligned}
$$

Note here that observations $\mathbf{r}_{\phi_1}$ and $\mathbf{r}_{\phi_2}$ cannot be assumed to be independent, since when comparing reward models across iterations and proxy reward models with the gold reward model, independence is not guaranteed.

This two-step procedure allows us to perform a detailed comparison of reward models going beyond the measurement of the mean gold reward.

### B.1  PROOFS

**Proposition B.1.** *Let $R_{\phi_1}, R_{\phi_2} \in \mathcal{R}$ be two reward functions and suppose they differ by an affine transformation, i.e., $R_{\phi_2} = a \cdot R_{\phi_1} + b$ for some $a \in \mathbb{R}^+$ and $b \in \mathbb{R}$. Then $R_{\phi_1'} = R_{\phi_2'}$, where $R_{\phi_i'} = \frac{1}{\sigma_i} \cdot (R_{\phi_i} - \mu_i)$ with $\sigma_i$ the standard deviation of $R_{\phi_i}$ and $\mu_i$ the mean.*

**Proof of Proposition B.1.**  First note that $R_2 = a' \cdot R_1' + b'$, with $a' = a \cdot \sigma_1 \in \mathbb{R}+$ and $b' = b + a \cdot \mu_1$. We have that $\mu_2 = \mathbb{E}(R_2) = b'$ and $\sigma_2 = a'$. Hence

$$R_2' = \frac{R_2 - \mu_2}{\sigma_2} \tag{4}$$

$$= \frac{R_2 - b'}{a'} \tag{5}$$

$$= \frac{a'R_1' + b' - b'}{a'} \tag{6}$$

$$= R_1'. \tag{7}$$

**Proposition B.2.** *Given i.i.d. observations $x_1, ..., x_n$ from random variable $x \sim \rho$, and a policy $\pi_\theta$, we have that observations of rewards $r_1, ..., r_n$, where $r_i = R_\phi(x_i, y_i)$ for a deterministic reward function $R_\phi$ and $y_i \sim \pi_\theta(\cdot|x_i)$ for $i = 1, ..., n$, are i.i.d. observations of a random variable we denote by $Z$.*

**Proof of Proposition B.2.** Given that $X_i$ are independent and identically distributed (i.i.d.) and that $Y_i \sim \pi(\cdot|X_i)$, we first show that $Y_i$ are i.i.d..

To determine if $Y_i$ are independent, we need to check if the joint distribution of any pair $(Y_i, Y_j)$ for $i \neq j$ factorizes into the product of their marginal distributions.

Since $X_i$ are i.i.d., we have:

$$P(X_i, X_j) = P(X_i)P(X_j) \text{ for } i \neq j.$$

Given $Y_i \sim \pi(\cdot \mid X_i)$, $Y_i$ and $Y_j$ are conditionally independent given $X_i, X_j$ for $i \neq j$ and the conditional distribution of $Y_i$ given $X_i$ is independent of $X_j$ for $j \neq i$, such that

$$P(Y_i, Y_j \mid X_i, X_j) = P(Y_i \mid X_i) P(Y_j \mid X_j)$$

Using the law of total probability, the joint distribution $P(Y_i, Y_j)$ can be written as

$$P(Y_i, Y_j) = \iint P(Y_i, Y_j \mid X_i, X_j) P(X_i, X_j) \, dX_i dX_j.$$

Substituting the factored form of the conditional and marginal distributions, we get

$$P(Y_i, Y_j) = \iint P(Y_i \mid X_i) P(Y_j \mid X_j) P(X_i) P(X_j) \, dX_i dX_j.$$

Since $P(X_i)$ and $P(X_j)$ are independent, this simplifies to

$$P(Y_i, Y_j) = \left( \int P(Y_i \mid X_i) P(X_i) \, dX_i \right) \times \left( \int P(Y_j \mid X_j) P(X_j) \, dX_j \right). \tag{8}$$

$$\tag{9}$$

This shows that
$$P(Y_i, Y_j) = P(Y_i) P(Y_j),$$
which means $Y_i$ and $Y_j$ are independent for $i \neq j$.

We now check if $Y_i$ are identically distributed. Since $Y_i \sim \pi(\cdot \mid X_i)$ and $X_i$ are i.i.d., the marginal distribution of $Y_i$ is obtained by marginalizing over $X_i$, which yields

$$P(Y_i = y) = \int P(Y_i = y \mid X_i = x) P(X_i = x) \, dx.$$

Given that $X_i$ are identically distributed, the distribution $P(X_i)$ is the same for all $i$. Therefore, the marginal distribution $P(Y_i)$ is the same for all $i$, indicating that $Y_i$ are identically distributed.

Now, given $R_i = r(X_i, Y_i)$ where $r$ is some deterministic function, we need to determine whether $R_i$ are i.i.d., given that $X_i$ are i.i.d. and $Y_i \sim \pi(\cdot \mid X_i)$.

Since $X_i$ are i.i.d., $X_i$ and $X_j$ are independent for $i \neq j$. We have established that $Y_i$ and $Y_j$ are also independent for $i \neq j$. Because $r$ is a deterministic function, $R_i$ is fully determined by $(X_i, Y_i)$. Specifically
$$R_i = r(X_i, Y_i) \text{ and } R_j = r(X_j, Y_j).$$

Given that $(X_i, Y_i)$ and $(X_j, Y_j)$ are independent pairs, it follows that $R_i$ and $R_j$ are also independent. This is because the independence of $(X_i, Y_i)$ and $(X_j, Y_j)$ implies that the mapping through $r$ does not introduce any new dependency between $R_i$ and $R_j$.

Next, we need to check if $R_i$ are identically distributed. Since $X_i$ are i.i.d. and $Y_i \sim p(\cdot \mid X_i)$, the distribution of $(X_i, Y_i)$ is the same for all $i$. The function $r$ is deterministic and applies the same transformation to each pair $(X_i, Y_i)$. Therefore, the distribution of $R_i = r(X_i, Y_i)$ will be the same for all $i$. This concludes the proof.

# C ADDITIONAL EXPERIMENTAL DETAILS

## C.1 HYPERPARAMETERS

Our hyperparameter settings mostly align with those used by the authors in Coste et al. (2024). The parameters for supervised fin-tuning are given in Table 1, reward model training hyperparameters are specified in Table 2, PPO parameters are given in Table 3, and the hyperparameters for synthesis with a policy are provided in Table 4.

Table 1: SFT hyperparameters.

| PARAMETER | VALUE |
|---|---|
| LEARNING RATE | $8e - 6$ |
| EPOCHS | 3 |
| BATCH SIZE | 4 |

Table 2: RM hyperparameters.

| PARAMETER | VALUE |
|---|---|
| LEARNING RATE | $1e - 5$ |
| EPOCHS | 5 |
| BATCH SIZE | 32 |

Table 3: PPO hyperparameters.

| PARAMETER | VALUE |
|---|---|
| LEARNING RATE | $1e - 6$ |
| COSINE ANNEALING SCHEDULER | $1e - 7$ |
| PPO STEPS | 6000 |
| BATCH SIZE | 32 |
| NUMBER OF ROLLOUTS | 256 |
| CHUNK SIZE | 32 |
| CLIPPING RANGE & VALUE | 0.2 |
| GAE LAMBDA | 0.95 |

## C.2 DATASET

We use the instructions and inputs contained in the popular AlpacaFarm dataset (Dubois et al., 2023; Taori et al., 2023). The entire dataset contains $52,000$ samples split into "sft" (10k), "preference" (20k), "unlabeled" (20k), and "val" (2k). We use the "val" split strictly only for validation. The instructions for the reward model training are sampled from the "preference" split and the instructions for PPO are sampled from the "unlabeled" split.

## C.3 PROMPT FORMAT

We follow the prompt format used in (Coste et al., 2024; Köpf et al., 2023), which is that of the v2 format used in Open Assistant. It uses special tokens `<|prompter|>` and `<|assistant|>`, and is consistent with the `GPTNeoXTokenizer` class.

To generate answers the model is prompted with the concatenation of instruction and input (if present), where inputs begin on a new line. The entire prompt begins with the special token `<|prompter|>` and ends with the end-of-text token `<|endoftext|>` to indicate the end of the instruction followed by the `<|assistant|>` token to start generating the answer.

In the case of the reward model the prompt should additionally contain an answer to the instruction, which is appended to the initial prompt and again ended with the `<|endoftext|>` token. Examples for both generation and reward modelling are given in Table 5.

Table 4: Generation hyperparameters.

| PARAMETER | VALUE |
|---|---|
| MAX INSTRUCTION LENGTH | 520 |
| MAX NEW TOKENS | 256 |
| PPO EPOCHS | 4 |
| TOP-P | 0.9 (1.0 FOR PPO) |
| TOP-K | 0 |
| TEMPERATURE | 1.0 |

Table 5: Example answer generation and reward modelling prompts with proper formatting.

| Answer generation prompt | Reward modelling prompt |
|---|---|
| `<|prompter|>Categorize the following items as either furniture or kitchen items.\nChair, Knife, Fork<|endoftext|> <|assistant|>` | `<|prompter|>Categorize the following items as either furniture or kitchen items.\nChair, Knife, Fork<|endoftext|> <|assistant|> Furniture:  Chair, Kitchen: Knife, Fork<|endoftext|>` |

### C.4 COMPUTATIONAL SETUP AND COST

All experiments were run on a single Nvidia A100. Running the full pipeline consisting of all 3 RLHF steps for 4 iterations takes approximately 35 hours per seed and configuration. Subsequently labelling the results with the 7B gold reward model takes approximately 18h when using an evaluation set of size 2000 and evaluating every 300 steps.

**Inference overhead for reward models.** Reward model ensembles and WCO incur an inference cost proportional to the number of models $K$ (in our case $K$ models must be evaluated). The number of models to be evaluated grows with each iteration. Weight averaging has the significant advantage of zero inference overhead compared to a single model, as the combination happens in parameter space.

**Cost performance trade-offs.** We summarise the trade-offs based on our results as follows:

- *Concat Data + From SFT (High Robustness, Moderate Cost Increase):* While the reward model training cost scales linearly for *Concat Data*, it is worth noting that RM training is often faster than the generation and PPO phases of the pipeline. Thus, the overhead of concatenating data is often negligible compared to the cost of a failed run.

- *Take Last + Ensemble/WCO (High Cost):* This has high inference costs during PPO (number of forward passes scales linearly with the iteration) but constant reward model training cost. We observed that for smaller models (70M), the computational overhead of ensembles does not yield relative gains over simple data aggregation. For larger reward models performance is more comparable.

- *Weight Averaging (High Efficiency):* This method has no inference overhead and standard training costs, i.e., it is the most efficient. However, our results show that it is less effective at mitigating overoptimisation than for example *Concat Data*.

Our findings suggest that the cheapest methods (like *Take Last*) often lead to policy collapse or stagnation. Therefore, spending the marginal extra compute on *Concat Data* is likely the most efficient option in the long-run.

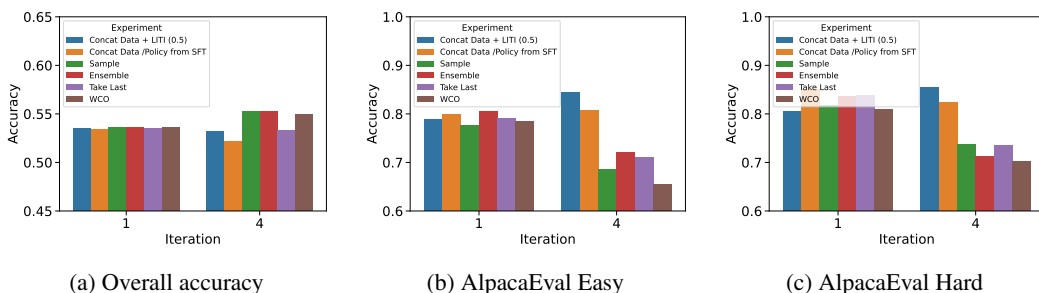

Figure 9: RewardBench performance for 70M reward models.

# D   FULL REWARD MODEL EVALUATION ON REWARDBENCH

We evaluate the reward models from the first and final iterations that obtained via the different design choices on the reward model benchmark RewardBench (Lambert et al., 2024). In particular, since we train on AlpacaFarm, we report the performance on the AlpacaEval Easy and Hard splits as well as the overall accuracy averaged across all subsets of the benchmark (Figure 9 and Figure 8).

We observe that the proxy reward models, which correspond to methods that ultimately achieve high gold reward in the final iteration, obtain higher or comparable accuracy on the AlpacaEval subsets when comparing the first and final iterations. This is the case for *Concat Data* with the 70 and 160 million reward models, as well as *LITI* with the larger reward model. Interestingly, also *LITI* with the smaller reward model results in performance gains between first and final iteration despite achieving a lower gold reward. We hypothesize that this is due to the concatenation of preference datasets for reward model training. The remaining design choices yield final reward models that achieve lower accuracy on AlpacaEval subsets than their counterparts trained in the initial iteration. The full results of the experiments are also reported in Table 6.

Table 6: RewardBench results grouped by design choice and reward model size.

| RM Size | Experiment | Iteration | Accuracy | Alpacaeval Hard | Alpacaeval Easy | Alpacaeval Length |
|---------|-----------|-----------|----------|-----------------|-----------------|-------------------|
| 70m | WCO | 4 | 0.550 | 0.702 | 0.656 | 0.588 |
|  |  | 1 | 0.536 | 0.809 | 0.786 | 0.617 |
|  | Take Last | 4 | 0.533 | 0.735 | 0.711 | 0.586 |
|  |  | 1 | 0.535 | 0.838 | 0.790 | 0.580 |
|  | Sample | 4 | 0.553 | 0.737 | 0.686 | 0.584 |
|  |  | 1 | 0.537 | 0.817 | 0.777 | 0.609 |
|  | Ensemble | 4 | 0.553 | 0.713 | 0.721 | 0.591 |
|  |  | 1 | 0.536 | 0.836 | 0.806 | 0.604 |
|  | Concat Data /Policy from SFT | 4 | 0.521 | 0.825 | 0.807 | 0.586 |
|  |  | 1 | 0.534 | 0.851 | 0.800 | 0.586 |
|  | Concat Data + LITI (0.5) | 4 | 0.532 | 0.855 | 0.845 | 0.592 |
|  |  | 1 | 0.536 | 0.805 | 0.789 | 0.584 |
| 160m | WCO | 4 | 0.497 | 0.809 | 0.784 | 0.576 |
|  |  | 1 | 0.439 | 0.943 | 0.889 | 0.534 |
|  | Sample Exclusive | 4 | 0.485 | 0.847 | 0.797 | 0.562 |
|  |  | 1 | 0.456 | 0.917 | 0.889 | 0.542 |
|  | Sample | 4 | 0.496 | 0.868 | 0.845 | 0.586 |
|  |  | 1 | 0.450 | 0.932 | 0.910 | 0.536 |
|  | Ensemble | 4 | 0.496 | 0.818 | 0.805 | 0.567 |
|  |  | 1 | 0.450 | 0.930 | 0.882 | 0.529 |
|  | Concat Data /Policy from SFT | 4 | 0.492 | 0.925 | 0.914 | 0.638 |
|  |  | 1 | 0.489 | 0.941 | 0.899 | 0.534 |
|  | Concat Data + LITI (0.5) | 4 | 0.482 | 0.928 | 0.922 | 0.632 |
|  |  | 1 | 0.466 | 0.918 | 0.886 | 0.566 |

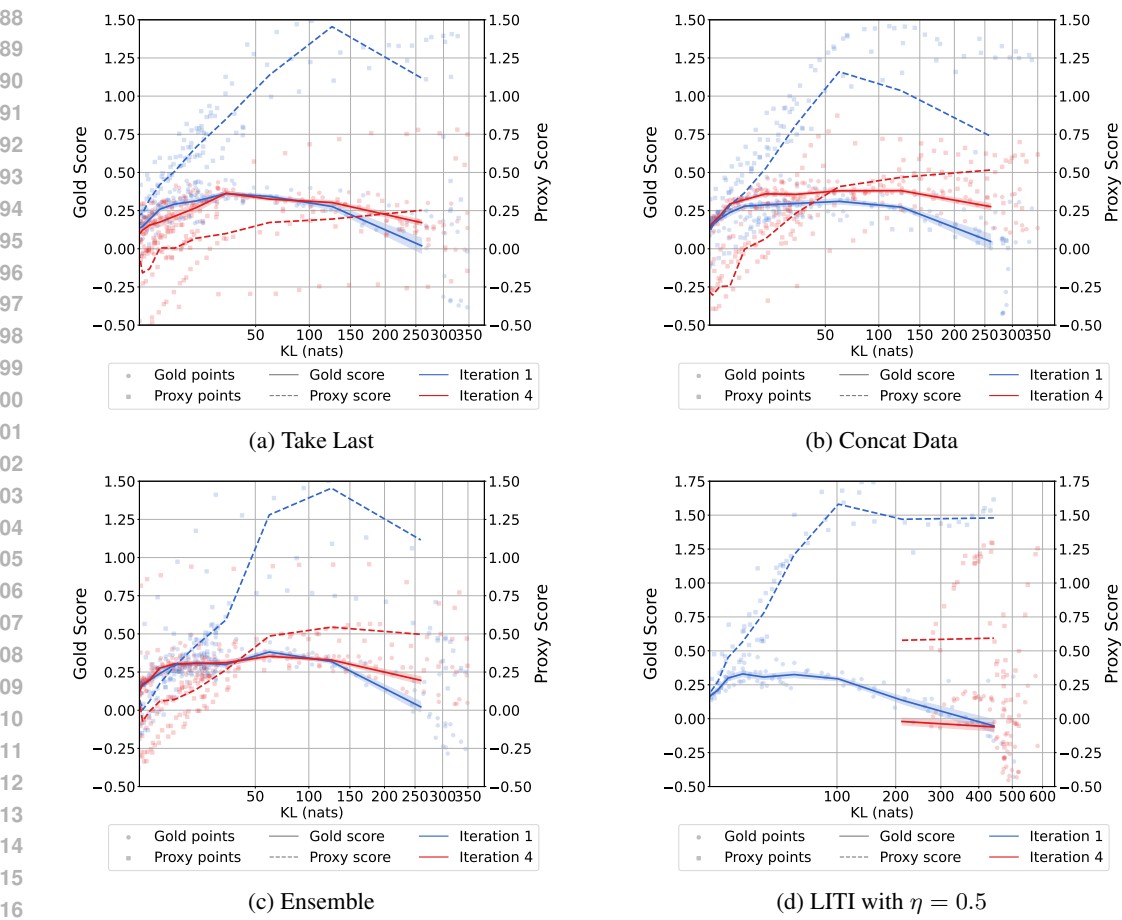

(a) Take Last

(b) Concat Data

(c) Ensemble

(d) LITI with $\eta = 0.5$

Figure 10: Ablation with $25\%$ label noise in the preference labels the $70m$ reward model is trained on. The hierarchy of design choices as well as the trends across iterations are consistent with the corresponding runs without label noise. Overall, all methods achieve lower gold reward than without label noise, which is expected.

## E    RESULTS WITH LABEL NOISE

To simulate noisy preference we conduct experiments with $25\%$ label noise. In particular, in the preference labelling phase, each preference label is flipped with probability $0.25$. This probability has been found to be empirically consistent with real-world data and has been used to study label noise in prior works (Coste et al., 2024). In Figure 10 we observe that the hierarchy among design choices as well as the trends across iterations are consistent with the results obtained without label noise. We still observe severe overoptimisation in the initial iteration, which some design choice can mitigate towards the final iteration. Additionally, all methods achieve lower gold reward, which is an expected effect of label noise.

# F    ADDITIONAL RESULTS

## F.1    CLOSING THE GAP BETWEEN PROXY AND GOLD REWARD FUNCTION

Here we provide additional experimental results for taking the last preference dataset and sampling the preference datasets with equal proportion. In terms of the rate at which the gap between proxy and gold reward functions is reduced over iterations, the sampling strategy (see Figure 11) falls in between concatenating all preference data and taking only the last dataset (see Figure 12).

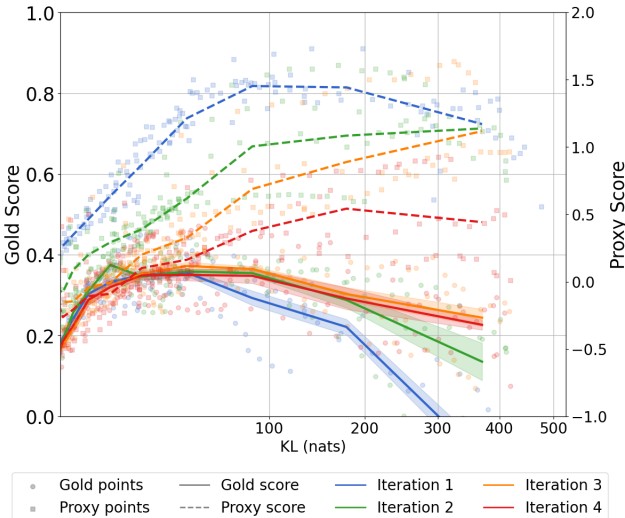

Figure 11: The gap between gold and proxy reward function when sampling from all preferences dataset equally to form the reward model training data.

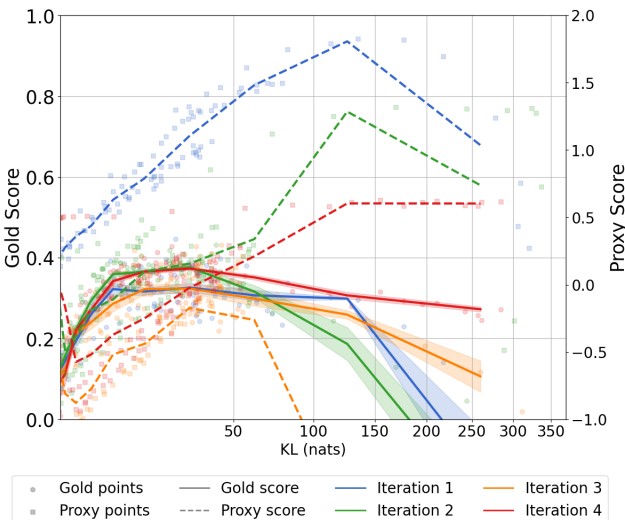

Figure 12: The gap between gold and proxy reward function when only taking the last preferences dataset for reward model training.

## F.2    ADDITIONAL RESULTS FOR COMBINING PREFERENCE DATA

In Figure 13  we provide the individual seeds for methods combining preference data across all iterations and in Figures 14 and 15 we provide the results for the sampling strategies. Figure 16 shows the MMD across iterations when only using the most recent preference dataset.

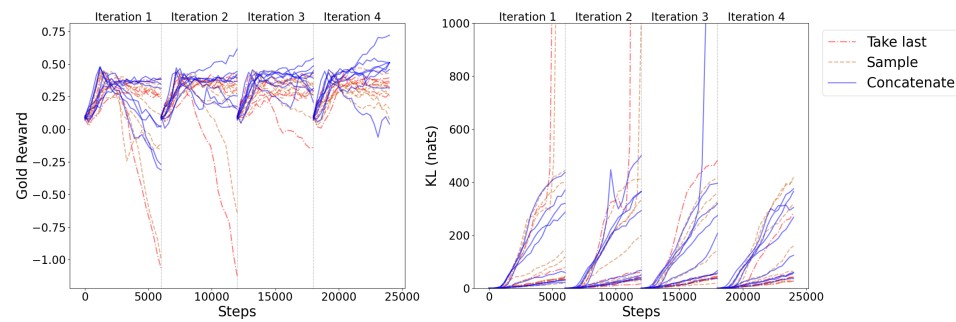

Figure 13: Gold score and KL of individual seeds across iterations for varying preference data combination methods.

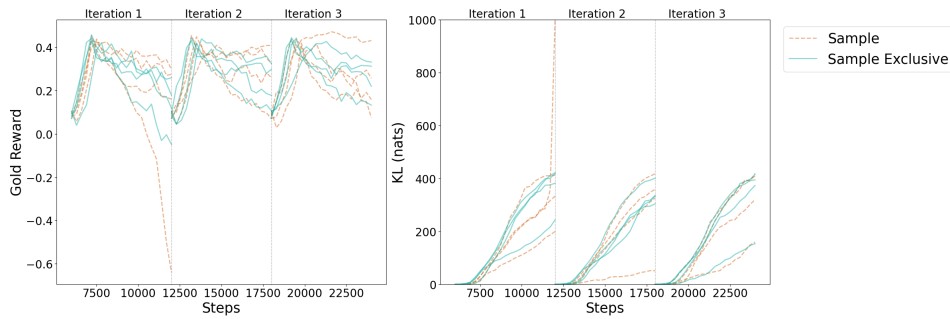

Figure 14: Gold score and KL of individual seeds across iterations comparing sampling with full coverage of the prompts vs random sampling.

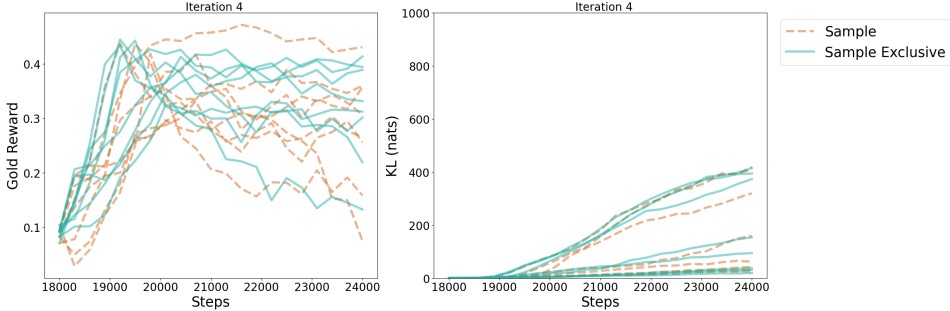

Figure 15: Gold score and KL of individual seeds in the fourth iteration comparing sampling with full coverage of the prompts vs random sampling.

### F.3 ADDITIONAL RESULTS FOR REWARD MODEL TRANSFER

Here we provide additional results for methods addressing reward model transfer. Figure 17 and 18 show the individual training seeds of the methods across iterations.

### F.4 ADDITIONAL RESULTS FOR POLICY INITIALISATION

Here we provide additional results for the policy initialisation methods (Figures 19 and 20). In particular, we plot the runs associated with each seed, highlighting seeds that are strongly overoptimised and can not be recovered by the respective methods.

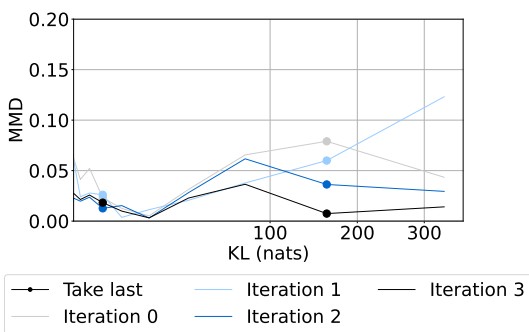

Figure 16: Taking the last preference dataset results in consistently low MMD, in the final iteration.

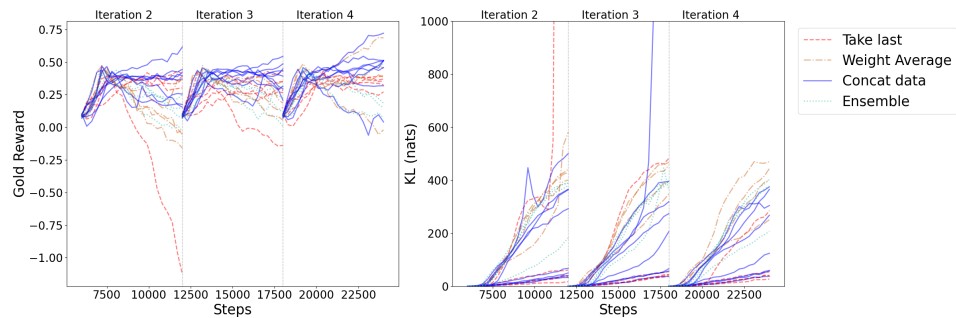

Figure 17: Gold score and KL of individual seeds across iterations comparing reward function choices.

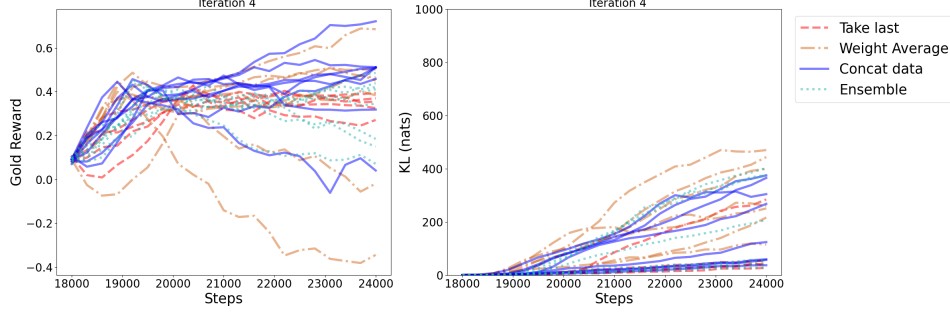

Figure 18: Gold score and KL of individual seeds in the fourth iteration comparing reward function choices.

## F.5  INVESTIGATING ENTROPY COLLAPSE

In Figure 21 we show the policy entropy throughout training for the *Take Last* policy initialisation method. The performance collapse extends beyond a simple entropy collapse, suggesting that the policy is exploiting weaknesses in the proxy reward model that cannot be corrected in subsequent iterations.

## F.6  ON TRAINING STABILITY ACROSS SEEDS AND ITERATIONS

As is common with RL fine-tuning, we observed variance across random seeds. To mitigate this, we have performed training with 8 random seeds (significantly more than what is standard in the literature) and report the average performance and standard errors. While we focused on the effect of different methods on overoptimisation, we also observed that the methods proposed, particularly those

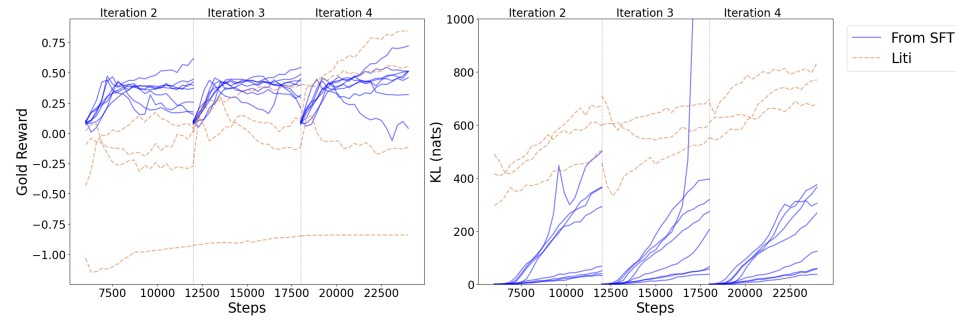

Figure 19: Gold score and KL of individual seeds across iterations comparing policy initialisation methods.

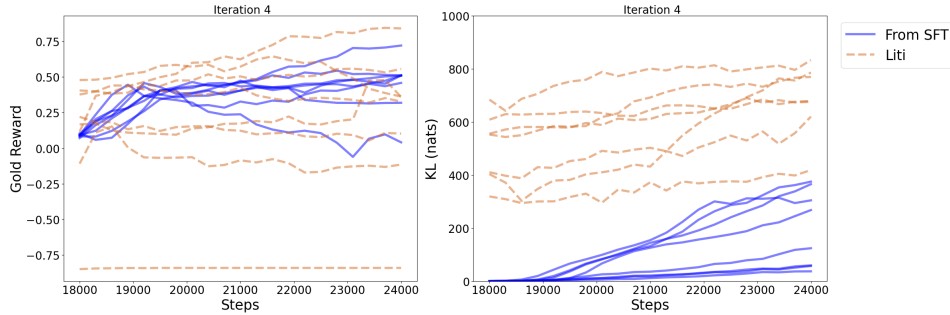

Figure 20: Gold score and KL of individual seeds in the final iteration comparing policy initialisation choices.

that reduce overoptimisation, tend to lead to more stable training. For instance, From SFT policy initialization consistently showed lower variance in performance compared to other initialization strategies, suggesting improved stability. Please find a summary of these results in Table 7

Table 7: Mean and standard deviation across seeds at the end of the fourth iteration.

| Method | Mean | Standard Deviation |
|---|---|---|
| Take last Data | 0.3572 | 0.0406 |
| Sample | 0.2761 | 0.0381 |
| Concat Data / Policy from SFT | 0.4477 | 0.0653 |
| Ensemble | 0.3136 | 0.0515 |
| Worst-Case Optimisation | 0.2942 | 0.0450 |
| Weight Average | 0.3035 | 0.1248 |
| Concat Data + LITI | 0.1991 | 0.1678 |
| Sample + Take last Policy | -0.0632 | 0.1055 |

## G  AN EXAMPLE OF IDIOSYNCRATIC BEHAVIOUR

Below we compare example responses from the *Take Last* and *From SFT* Policy initialisation methods at the start of training and at the end of the fourth iteration. The response at the end of training obtained with *Take Last* initialisation is notably more repetitive indicating severe overoptimisation. On the other hand, while still slightly repetitive, the *From SFT* initialisation results in a more sensible and less idiosyncratic response.

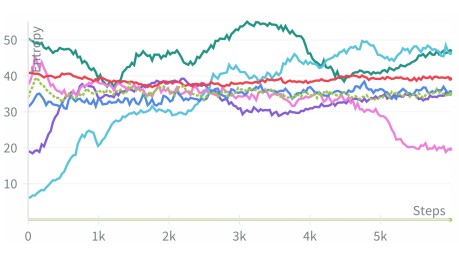

(a) All seeds final iteration.

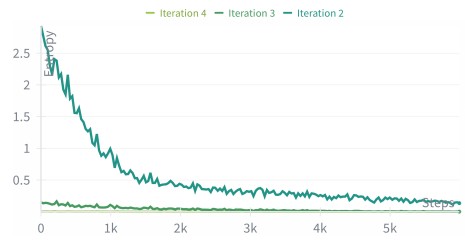

(b) Entropy collapse across iterations.

Figure 21: We plot the entropy of the policy against training steps for the *Take Last* policy initialisation strategy. In the final iteration only one out of eight seeds has collapsed.

---

**_Take Last_ Initialisation**

**Instruction:**
"Given the following input, construct a creative story.
A magic bow and arrow"

**Response from $\pi^{sft}$:**
"The Arrow of time was a powerful magical weapon that could grant the user special abilities. It could grant them the ability to change shape, fly, and even beamed down with sound. The Arrow of time could also be used to travel to distant lands, granting the user access to magical items and secrets."

**Response at the end of iteration $4$:**
"The story demonstrates the power and ability the ability to use the creativity and the practice in the practice and the practice in the creativity. This example demonstrates the ability to practice the practice in the practice in the practice. This demonstrates the practice in the practice in the practice in the practice. This demonstrates the practice in the practice in the practice in the practice. This demonstrates the practice in the practice in the practice in the practice. This demonstrates the practice in the practice in the practice in the practice. This demonstrates the practice in the practice in the practice in the practice. This demonstrates the practice in the practice in the practice in the practice in the practice. This demonstrates the practice in the practice in the practice in the practice. This demonstrates the practice in the practice in the practice in the practice. This demonstrates the practice in the practice in the practice in practice. This demonstrates the practice in the practice in the practice in the practice. This demonstrates the practice in the practice in the practice. This demonstrates the practice in the practice in the practice. This demonstrates the practice in the practice in the practice. This demonstrates the practice in the practice in the practice in the practice. This demonstrates the practice in the practice in the practice in the practice. This demonstrates the practice in the practice in the practice in the practice. This demonstrates"

---

*Concat Data* and *From SFT* Initialisation

**Instruction:**
"Given the following input, construct a creative story.
A magic bow and arrow"

**Response from $\pi^{sft}$:**
"The Arrow of time was a powerful magical weapon that could grant the user special abilities. It could grant them the ability to change shape, fly, and even beamed down with sound. The Arrow of time could also be used to travel to distant lands, granting the user access to magical items and secrets."

**Response at the end of iteration $4$:**
"A young man named Tom had always wanted to explore the world, but he was scared of the unknown. He was determined to learn to use the skills and knowledge he had acquired in the outdoors and to create his own special abilities and abilities. He set off on an adventurous journey and soon found himself on a magical journey to the far away and mystical world of the Grimm. With the help of a mysterious young woman and a powerful and powerful magic, Tom learned to use the bow and arrow and to create his own unique and powerful abilities. He learned to use the skill and knowledge he had gained in the outdoors and to use it to help protect and help others. This story will inspire and engage readers and will help them to explore and develop their creativity and knowledge in the world of the imagination."

