# OpenReview forum: "Reward Model Overoptimisation in Iterated RLHF"
_ICLR.cc/2026/Conference — Submitted to ICLR 2026_

### Official Review · Reviewer_C9FL · 2025-10-22

**Soundness:** 3
**Presentation:** 3
**Contribution:** 2
**Rating:** 6
**Confidence:** 4

**Summary:**

This paper investigates how to best mitigate reward model overoptimization in iterated RLHF, where models are repeatedly fine-tuned on new preference data. The authors systematically analyze three critical design choices: how to manage preference data across iterations, how to formulate the reward function, and how to initialize the policy for each training cycle. The study concludes that the most robust and effective strategy is to concatenate all preference data from previous rounds and re-initialize the policy from its original supervised fine-tuned (SFT) checkpoint before each new iteration.

**Strengths:**

1. This paper systematically tests key design choices on reward computation, preference data collection and model initialization, providing a clear, structured understanding of how different components affect training stability and performance.
2. The paper is well-written with good presentation and analysis
3. The use of Maximum Mean Discrepancy (MMD) to measure the distributional differences between the proxy and the gold reward models helps explain the effect distribution shift. This offers a more nuanced view of overoptimization, revealing how the reward distributions diverge, especially at high KL values

**Weaknesses:**

My main concern is the model size—both for the language model and reward model. It's unclear whether these findings transfer to larger reward models, though intuitively the analysis should hold even with more capable models.

The work also relies heavily on the assumption of a good, static "gold" reward model. In practice, such a model rarely exists for most post-training tasks. It will evolve over time and carry its own misalignment with human preferences.

**Questions:**

1. The `Sample + Take last Policy` The strategy shows a dramatic performance collapse, which it never recovers from. Was this failure due to entropy collapse, or did the policy learn specific, unrecoverable adversarial behaviors that the updated reward model couldn't penalize effectively?

2. Do you hypothesize there is a model scale at which LITI would consistently outperform the robust `From SFT` strategy? If so, would it be because a more capable reward model provides better-calibrated gradients, making it easier for the policy to "unlearn" overoptimized behaviors from the previous step?

3. For a fixed compute budget, is it better to run more iterations with a cheaper data strategy (like `Take last Data`) or fewer, more expensive iterations using the full `Concat Data` approach? The results suggest `Concat Data` is superior, but how does that trade off with the number of iterations one can afford?

---

> ### Author Response · Authors · 2025-11-26
> **Official Comment by Authors**
>
> We appreciate the Reviewer’s efforts and insightful comments and are grateful for the positive evaluation. We now address the raised concerns and questions in detail.
>
> ## Weaknesses
>
> > Weakness 1) My main concern is the model size—both for the language model and reward model. It's unclear whether these findings transfer to larger reward models, though intuitively the analysis should hold even with more capable models.
>
>  As noted by the Reviewer, the analysis is expected to hold for larger policy and reward models. This is supported by the findings of [1], which show that policy size does not play a significant role in the overoptimization dynamics and larger reward models can be still overoptimized (though at a slower rate).
>
> > Weakness 2) The work also relies heavily on the assumption of a good, static "gold" reward model. In practice, such a model rarely exists for most post-training tasks. It will evolve over time and carry its own misalignment with human preferences.
>
> We agree with the Reviewer that in practice human preferences may drift over time. While certainly relevant and an important problem, this is beyond the scope of this paper. **In our work we are specifically investigating the fundamental problem of overoptimization, and not tackling/accounting for human preference drift does not detract from the contributions of our work**. We see our work as a direct step towards the more complex scenario with shifting reward functions. **It is also important to note that, unfortunately, studying overoptimization with real human data would be extremely costly (especially in the academic setting setting), even more so for iterative RLHF, which requires policy preferences in each iteration**. For this reason, in research it is only feasible (also research at leading industry labs [1]) to simulate human preferences via a gold reward model [1,2]). **We have revised section 3.1 and the limitations section based on this discussion.**
>
> ## Questions
>
> > Question 1) The `Sample + Take last Policy` strategy shows a dramatic performance collapse, which it never recovers from. Was this failure due to entropy collapse, or did the policy learn specific, unrecoverable adversarial behaviors that the updated reward model couldn't penalize effectively?
>
> We only observed entropy collapse in ⅛ seeds, where half way into the second iteration the entropy had fully collapsed. However, our results suggest that this is not the primary driver of poor performance. The policies that continue to optimize proxy reward, however, do not obtain any gains in terms of gold reward. It is primarily due to the specific behaviours that exploit weaknesses in the proxy reward models. This interpretation is supported by the observation that the failure modes correspond to narrow, repeated action sequences rather than a uniform loss of exploration or stochasticity (i.e., entropy does not collapse for most seeds). **We included an example of this behaviour in comparison to the *From SFT* initialisation in Appendix G of the revised manuscript. We have added an exemplary plot of the entropy across different runs in Appendix F.5. We updated the corresponding second paragraph of Section 5.4 accordingly.**
>
> > Question 2) Do you hypothesize there is a model scale at which LITI would consistently outperform the robust `From SFT` strategy?…
>
> Yes, we hypothesise that *LITI* could outperform *From SFT* at larger reward model scales. Experiments with a larger reward model show that LITI benefits substantially from increased reward model capacity. This is likely due to two complementary effects: 1) better-calibrated gradients that support recovery from overoptimisation; and 2) larger reward models have been shown to overoptimise less aggressively [1], which should lead to safer intermediate policies and more stable interpolation paths. Thus, as the reward model scales, *LITI* may become increasingly competitive, while *From SFT* remains limited by repeated initialisation from the SFT checkpoint. **We have revised paragraphs 2 and 3 of section 5.4 accordingly.**

---

> > ### Author Response · Authors · 2025-11-26
> > **Official Comment by Authors**
> >
> > > Question 3) For a fixed compute budget, is it better to run more iterations with a cheaper data strategy (like `Take last Data`) or fewer, more expensive iterations using the full `Concat Data` approach?...
> >
> > Indeed, based on our results, one should prioritize running fewer, more expensive iterations using the *Concat Data* approach rather than extending a cheaper strategy like *Take Last* over many iterations. Specifically, two iterations of *Concat Data* achieve a higher gold score than four iterations with the *Take Last* strategy. In fact, the two additional iterations required for *Take Last* to achieve similar performance, are significantly more costly than running the *Concat Data* strategy for fewer iterations. However, for the larger reward model, we did observe stronger performance from the *Ensemble* and *WCO* strategies that only train on the most recent preference dataset in each iteration. **We have revised the conclusion accordingly.**
> >
> > We thank the Reviewer again for their positive evaluation and valuable feedback. If the Reviewer is satisfied with our responses, we would be grateful if they would kindly consider increasing their score.
> >
> > Sincerely,
> >
> > The Authors
> >
> > [1] Gao, L., Schulman, J., and Hilton, J. Scaling Laws for Reward Model Overoptimization. In *ICML 2023*.
> >
> > [2] Coste, T., Anwar, U., Kirk, R., & Krueger, D. Reward Model Ensembles Help Mitigate Overoptimization. In *ICLR 2024*.

---

> > > ### Comment · Reviewer_C9FL · 2025-11-26
> > > **Response by Reviewer**
> > >
> > > Thanks for providing clarification. I will maintain my scoring.

---

### Official Review · Reviewer_utsC · 2025-10-31

**Soundness:** 2
**Presentation:** 3
**Contribution:** 3
**Rating:** 4
**Confidence:** 5

**Summary:**

This paper investigates reward-model over-optimization in *iterated RLHF*: within the iterative loop—preference data collection → reward model training→ Policy optimazation—it examines which design choices mitigate or amplify over-optimization. The authors run a comparative evaluation between a *gold reward* and a *proxy reward* on *AlpacaFarm* and report key findings for iterated RLHF.

**Strengths:**

1. Provides a detailed study of reward over-optimization, factorizing iterated RLHF into three stages and empirically exploring actionable components in each stage.
2. Introduces metrics such as *MMD* and *KL–reward curves* to analyze over-optimization phenomena.
3. Delivers thorough experimental analyses; the conclusions are insightful and offer practical guidance for related applications.

**Weaknesses:**

1. Beyond proximity to the gold reward, the paper should report testset metrics (e.g., pairwise accuracy) for the proxy reward across iterations to provide more comparable evidence.
2. Although the gold and proxy rewards differ substantially in parameter count, report their performance  on held-out test sets and on public benchmarks (e.g., RewardBench) may lead resuslt more clear.
3. Conclusions drawn from a single dataset may be biased; the paper should evaluate on more datasets and base models to assess generalization. It is also advisable to consider a stronger LLM-as-judge as the gold reward to provide a more robust supervision signal.

If the authors can provide additional experiments to address these concerns, I would be happy to raise my score.

**Questions:**

See above weaknesses

---

> ### Author Response · Authors · 2025-11-26
> **Official Comment by Authors**
>
> We appreciate the Reviewer’s efforts and valuable comments on our work. We are confident that we can address the weaknesses raised by the Reviewer. We respectfully ask that you consider adjusting the score in light of our detailed clarifications and additional results.
>
> > Weaknesses 1 & 2) Test set metrics and reward model performance on public benchmarks such as RewardBench.
>
> We respectfully disagree with the assertion that proximity to the gold reward is an insufficient evaluation metric. **Gold-reward performance, as well as the proximity of proxy reward models to the gold reward model, is the central metric for evaluating and quantifying overoptimization. Our evaluation methodology closely follows [1] and [2] and extends it via the MMD metric. We also emphasize that all results reported in the manuscript are computed on the AlpacaFarm test split, which is never used during training.**
>
> **In response to the Reviewer’s request, we have additionally evaluated the reward models on RewardBench [3]. We have included this evaluation in Section 5.5 of the revised version and provide additional results in Appendix D (also table below for convenience).**
>
> In summary, proxy reward models, corresponding to methods achieving high gold reward in the final iteration, achieve comparable or better performance on AlpacaEval subsets in the final iteration (Concat Data 70m, LITI 160M, Concat Data 160m) compared to the first iteration. For other methods the reward models in the final iteration achieve lower accuracy on AlpacaEval subsets than those trained in the initial iteration.
>
> While we are pleased to provide this supplementary evaluation, we note that RewardBench performance is less diagnostic for our setting than gold-reward alignment: due to distribution shift, reward models in later iterations may effectively mitigate overoptimization even if their accuracy on the RewardBench test set is lower.
>
> **Table 1) RewardBench results.**
>
> | RM Size | Experiment | Iteration | Accuracy | AlpacaEval Hard | AlpacaEval Easy | AlpacaEval Length |
> |--------|---|--|---|--|--|----|
> | 70m    | WCO  | 4 | 0.550    | 0.702  | 0.656 | 0.588   |
> | 70m    | WCO | 1 | 0.536    | 0.809 | 0.786  | 0.617 |
> | 70m    | Take Last| 4 | 0.533    | 0.735 | 0.711 | 0.586 |
> | 70m    | Take Last | 1 | 0.535    | 0.838 | 0.790 | 0.580 |
> | 70m    | Sample | 4 | 0.553    | 0.737 | 0.686 | 0.584  |
> | 70m    | Sample | 1 | 0.537    | 0.817 | 0.777 | 0.609 |
> | 70m    | Ensemble  | 4 | 0.553    | 0.713 | 0.721| 0.591 |
> | 70m    | Ensemble  | 1 | 0.536    | 0.836 | 0.806 | 0.604 |
> | 70m    | Concat Data / Policy from SFT | 4 | 0.521    | 0.825 | 0.807 | 0.586  |
> | 70m    | Concat Data / Policy from SFT | 1 | 0.534    | 0.851 | 0.800 | 0.586  |
> | 70m    | Concat Data + LITI (0.5) | 4 | 0.532    | 0.855 | 0.845  | 0.592   |
> | 70m    | Concat Data + LITI (0.5) | 1 | 0.536  | 0.805 | 0.789 | 0.584 |
> | 160m   | WCO | 4 | 0.497| 0.809 | 0.784  | 0.576 |
> | 160m   | WCO | 1 | 0.439 | 0.943 | 0.889| 0.534  |
> | 160m   | Sample Exclusive  | 4| 0.485 | 0.847 | 0.797 | 0.562 |
> | 160m   | Sample Exclusive | 1  | 0.456    | 0.917 | 0.889 | 0.542 |
> | 160m   | Sample | 4  | 0.496  | 0.868 | 0.845 | 0.586  |
> | 160m   | Sample | 1| 0.450 | 0.932| 0.910 | 0.536  |
> | 160m   | Ensemble   | 4  | 0.496 | 0.818 | 0.805 | 0.567 |
> | 160m   | Ensemble | 1 | 0.450 | 0.930 | 0.882 | 0.529 |
> | 160m   | Concat Data / Policy from SFT    | 4 | 0.492    | 0.925| 0.914 | 0.638 |
> | 160m   | Concat Data / Policy from SFT    | 1 | 0.489    | 0.941 | 0.899 | 0.534 |
> | 160m   | Concat Data + LITI (0.5)| 4  | 0.482    | 0.928  | 0.922 | 0.632   |
> | 160m   | Concat Data + LITI (0.5  | 1 | 0.466    | 0.918  | 0.886  | 0.566    |

---

> > ### Author Response · Authors · 2025-11-26
> > **Official Comment by Authors**
> >
> > > Weakness 3) Conclusions drawn from a single dataset may be biased;... It is also advisable to consider a stronger LLM-as-judge as the gold reward to provide a more robust supervision signal.
> >
> > While we agree that in principle additional datasets would strengthen the experimental section, **our evaluation on AlpacaFarm provides sufficiently strong evidence for the claims made in our paper and is an evaluation consistent with the literature ([1], [2], also [4])**. In particular, AlpacaFarm is a widely used benchmark for controlled studies in RLHF, specifically designed for systematic investigations and has been used as a single dataset in prior work in the field (e.g. [1]). **The size of the gold reward model (7B) is fully consistent with prior works ([1] and [2]). The important criteria is its relative size in comparison to the proxy reward model, such that 7B is sufficiently large.**
> >
> > We would like to emphasize that running the entire RLHF pipeline, that is, preference data generation, labelling, reward model training, policy optimization and that for multiple iterations is a large scale evaluation. Consequently, we focused our resources on a thorough evaluation of the critical design choices and reward models at the core of this pipeline.
> >
> > Sincerely,
> >
> > The Authors
> >
> > [1] Coste, T., Anwar, U., Kirk, R., & Krueger, D. Reward Model Ensembles Help Mitigate Overoptimization. In *ICLR 2024*.
> >
> > [2] Gao, L., Schulman, J., & Hilton, J. Scaling Laws for Reward Model Overoptimization. In *ICML 2023*
> >
> > [3] Lambert, N. et al. RewardBench: Evaluating Reward Models for Language Modeling. *arXiv preprint arXiv:2403*.13787, 2024.
> >
> > [4] Zhu, B., Jordan, M., and Jiao, J. Iterative Data Smoothing: Mitigating Reward Overfitting and Overoptimization in RLHF. In *ICML 2024*.

---

### Official Review · Reviewer_JWGj · 2025-10-31

**Soundness:** 3
**Presentation:** 2
**Contribution:** 3
**Rating:** 4
**Confidence:** 4

**Summary:**

This paper studies reward model overoptimization in iterated RLHF. The authors define three key levers in the loop—(i) how preference data from prior rounds are reused, (ii) how to form the reward signal from multiple trained RMs, and (iii) policy initialization at each round—and evaluate their impact in a controlled AlpacaFarm setup with a “gold” RM as a stand-in for human labels. Methodologically, they also propose using distributional comparisons (MMD) between proxy and gold rewards (rather than only means) along the KL–reward curve during PPO training. Main findings: (1) overoptimization tends to decline across iterations as the proxy RM better matches the gold RM on-policy; (2) concatenating preference data across rounds is the most reliable way to curb overoptimization; (3) restarting from the SFT policy every round is the most robust (though less flexible) initialization; (4) combining RMs via ensembles/weight averaging gives modest gains with efficiency trade-offs; and (5) benefits plateau after ≈3 iterations, and some overoptimization remains even after four.

**Strengths:**

Well-scoped, decision-oriented study. The three knobs cover the practical choices teams actually debate; the recommendations are specific and replicable.

Concatenating preference data clearly helps. Strong and consistent gains vs. take-last/sample, especially in mid-KL regions where overoptimization tends to bite.

Policy resets matter. From-SFT avoids “digging the hole deeper”; recovering from an overoptimized policy is empirically hard—even with later iterations.

Distributional metric. The MMD view surfaces high-KL divergence that average scores miss, showing when iteration helps and when exploitation resumes.

Scale sensitivity checks. Showing that larger RMs (160M vs 70M) unlock more benefit for ensembling/WCO gives a plausible path as capacity grows.

**Weaknesses:**

Gold-RM surrogate limits external validity. A single fixed “gold” RM (and one dataset) can imprint its biases; real human-in-the-loop dynamics might differ (drift, noise, inconsistency).

Narrow task/model scope. Pythia-410M policies and 70M/160M RMs on AlpacaFarm only; conclusions might shift with stronger instruction-tuned policies, adversarial prompts, or safety domains.

Compute accounting is thin. We don’t see wall-clock/GPU hours per iteration/choice, nor inference overhead for ensembles/WCO vs. weight-averaging.

Initialization trade-offs underexplored. From-SFT is robust but may cap upside; LITI can improve with larger RMs, but guidance on safe settings (η, early-stop) is light.

Theory is mostly appendix-level and aspirational. The performative-prediction framing is helpful but doesn’t yield predictive thresholds for when iteration ceases to help.

**Questions:**

Cost vs. benefit. Can you report GPU hours/tokens/s per iteration for (a) concat-data + take-last RM + From-SFT, (b) concat-data + ensemble, and (c) WCO and weight-avg? Practitioners need Pareto curves (gold score vs. cost).

Human variance. Any pilot with noisy/biased “gold” RMs (or a mixture of RMs) to emulate annotator disagreement? How do the design choices change under label noise or drift?

When not to reset. Are there regimes (larger RMs, conservative KL schedules, early-stop) where LITI reliably beats From-SFT after >4 iterations without catastrophic KL compounding? Concrete η schedules would help.

Beyond AlpacaFarm. Do the rankings (concat > sample ≈ last; From-SFT best) hold on a safety/toxicity or factuality workload with verifiable metrics, or under length-bias stress tests?

RM calibration. Do ensembling/weight-avg improve calibration (e.g., STARC, ECE/Brier of pairwise prefs) or only gold-score means?

MMD operationalization. How sensitive are your MMD conclusions to kernel choice/bandwidth and to KL bucketing? Could you release a minimal script so others can reproduce the curves on their pipelines?

---

> ### Author Response · Authors · 2025-11-26
> **Official Comment by Authors**
>
> We thank the reviewer for their efforts and for their feedback on our work. We are confident that we can address the raised concerns and kindly ask you to consider increasing the score based on our detailed explanation and additional results.
>
> ## Weaknesses
>
> > 1) Gold-RM surrogate limits external validity. A single fixed “gold” RM (and one dataset) can imprint its biases; real human-in-the-loop dynamics might differ (drift, noise, inconsistency).
>
> The use of a static fixed gold reward model is the standard approach for investigating reward model overoptimization in RLHF since real human labels are too costly and impractical [1,2,3,4]. The simulation of human preferences provides a controlled setting to systematically study the fundamental dynamics of iterated RLHF and isolate the impact of different design choices.
>
> **In order to address the reviewers concern, we have conducted experiments with $25$% label noise in the preference data. The results can be found in Appendix E of the revised manuscript.** For a more detailed discussion on these results as well as preference drift, please refer to our response to Question 2 below. We also note that Coste et al. [2] find that introducing preference noise slightly reduces overoptimization in the standard RLHF setting. However, overoptimization poses a risk with and without label noise.
>
> > 2. Narrow task/model scope. Pythia-410M policies and 70M/160M RMs on AlpacaFarm only; conclusions might shift with stronger instruction-tuned policies, adversarial prompts, or safety domains.
>
> Our evaluation pipeline is fully consistent with prior work [2] based on this dataset. We use similarly sized policies and reward models [1,2,3,4]. Importantly, [1] has found that policy size does not drive reward model overoptimization, hence an evaluation with larger policies would not bring meaningful new insights. Additionally, the effect of reward model size on overoptimization is smooth, meaning that our findings are expected to apply also for larger reward models.
>
> > 3. Compute accounting is thin. We don’t see wall-clock/GPU hours per iteration/choice, nor inference overhead for ensembles/WCO vs. weight-averaging.
>
> **We clarify the computational cost and we have revised the manuscript accordingly.**
> * *Training Cost*: As noted in Appendix C.4, running the full pipeline (3 steps) for 4 iterations takes approximately 35 hours per seed on a single Nvidia A100. The "Concatenate Data" strategy scales linearly in reward model training time as the dataset grows, whereas "Sample" or "Take Last" keeps RM training time constant.
> * *Inference Overhead*:
>   * *Ensembles/WCO*: These incur an inference cost proportional to the number of models $K$ (in our case $K$ models must be evaluated). The number of models to be evaluated grows with each iteration.
>   * *Weight Averaging*: This has the significant advantage of zero inference overhead compared to a single model, as the combination happens in parameter space.
>
> > 4. Initialization trade-offs underexplored. From-SFT is robust but may cap upside; LITI can improve with larger RMs, but guidance on safe settings (η, early-stop) is light.
>
> If the Reviewer could please clarify the specific concern, we would greatly appreciate it. We agree that there is a trade-off. Our results show that "From SFT" is indeed the most robust method because it resets the policy drift at every iteration. However, we explicitly acknowledge that this "limits optimization flexibility". Regarding LITI, our experiments show it is sensitive and struggles to recover from early overoptimization with smaller reward models. **However, we observe that LITI becomes viable for larger reward models**. As shown in Figure 6, LITI benefits substantially from increased reward model capacity (160M vs 70M), likely because larger RMs provide better-calibrated gradients. In the revised version of the paper we have clarified that LITI is recommended only when using sufficiently expressive reward models, while "From SFT" remains the safer default.
>
> > 5. Theory is mostly appendix-level and aspirational. The performative-prediction framing is helpful but doesn’t yield predictive thresholds for when iteration ceases to help.
>
> **We have included a rigorous theoretical framework in the Appendix precisely because its primary role is to provide a rigorous framing for why the empirical mitigations work, rather than to derive predictive thresholds.** Specifically, the theory explains that data aggregation works by reducing estimation variance (Prop A.3) , ensembles reduce systematic error (Prop A.5) , and resetting bounds policy drift (Prop A.6). These theoretical insights directly support our empirical findings, even if the strict conditions are not met in deep RL settings.

---

> > ### Author Response · Authors · 2025-11-26
> > **Official Comment by Authors**
> >
> > ## Questions
> > > 1. Cost vs. benefit. Can you report GPU hours/tokens/s per iteration for (a) concat-data + take-last RM + From-SFT, (b) concat-data + ensemble, and (c) WCO and weight-avg? Practitioners need Pareto curves (gold score vs. cost).
> >
> > We agree that the derivation of computational costs is an important aspect. However, given the large scale of the study and the limited computational resources at our institution we were not able to re-run all the experiments in order to obtain the necessary performance summaries for deriving the pareto-plot suggested by the reviewer. We would like to also emphasize that the goal of this work was the systematic study of reward model overoptimisation.
> >
> > Based on our results, we summarise the trade-offs as follows:
> > * **Concat Data + From SFT (High Robustness, Moderate Cost Increase):** While the reward model training cost scales linearly for *Concat Data*, it is worth noting that RM training is often faster than the generation and PPO phases of the pipeline. Thus, the overhead of concatenating data is often negligible compared to the cost of a failed run.
> > * **Take Last + Ensemble/WCO (High Cost):** This has high inference costs during PPO (number of forward passes scales linearly with the iteration) but constant reward model training cost. We observed that for smaller models (70M), the computational overhead of ensembles does not yield relative gains over simple data aggregation. For larger reward models performance is more comparable.
> > * **Weight Averaging (High Efficiency):** This method has no inference overhead and standard training costs, i.e., it is the most efficient. However, our results show that it is less effective at mitigating overoptimization than for example *Concat Data*.
> > Our findings suggest that the cheapest methods (like *Take Last*) often lead to policy collapse or stagnation. Therefore, spending the marginal extra compute on *Concat Data* is likely the most efficient option in the long-run. **We have included this discussion in Appendix C.4.**
> >
> > > 2. Human variance. Any pilot with noisy/biased “gold” RMs (or a mixture of RMs) to emulate annotator disagreement? How do the design choices change under label noise or drift?
> >
> > **We have conducted additional experiments with $25$% label noise, i.e., each preference label obtained from the gold reward model is flipped with probability $0.25$. The new results can be found in Appendix E (Figure 10) of the revised manuscript. The evaluated design choices show a consistent hierarchy as well as trends across iterations with the results obtained without label noise.** Note that due to computational constraints and the short time frame, we evaluated a smaller sample of design choices.
> >
> > We agree that preference drift is an interesting and relevant problem, though it is beyond the scope of this paper. **We specifically focus on investigating the fundamental problem of overoptimization, its relevance to iterated RLHF, and how critical design choices can mitigate it.**  Not tackling/accounting for human preference drift does not detract from the contributions of our work.
> >
> > > 3. When not to reset. Are there regimes (larger RMs, conservative KL schedules, early-stop) where LITI reliably beats From-SFT after >4 iterations without catastrophic KL compounding? Concrete η schedules would help.
> >
> > Yes, based on our results the regime appears to be that of reward model scale. Our results in Figure 6 demonstrate that LITI improves significantly when moving from 70M to 160M reward models, whereas "From SFT" stays relatively stable. This suggests that with sufficiently large RMs, LITI may outperform "From SFT" by avoiding catastrophic forgetting of the previous iteration's gains, provided the RM is robust enough to guide the interpolation.
> >
> > > 4. Beyond AlpacaFarm. Do the rankings (concat > sample ≈ last; From-SFT best) hold on a safety/toxicity or factuality workload with verifiable metrics, or under length-bias stress tests?
> >
> > We believe the ranking **Concat > Sample ≈ Last and From-SFT > LITI** holds for domains where the primary challenge is distribution shift and reward hacking. The mechanism of *From SFT*, that is, resetting the policy to a known safe distribution, is domain-agnostic. However, in safety-critical domains, where "refusal" is a key feature, the dynamics might differ if the reward model penalizes helpfulness too strongly. **We acknowledge this limitation, but note that AlpacaFarm is the most popular dataset for this type of controlled methodological study [2] and influential papers in this area are based on it.**

---

> > > ### Author Response · Authors · 2025-11-26
> > > **Official Comment by Authors**
> > >
> > > > 5. RM calibration. Do ensembling/weight-avg improve calibration (e.g., STARC, ECE/Brier of pairwise prefs) or only gold-score means?
> > >
> > > Our primary metric is the gold score (ground truth performance) and MMD (distributional distance). We do not explicitly measure ECE. However, MMD serves as a proxy for calibration in the high-reward tail. Figure 5c shows that Weight Averaging and Ensembling do not outperform the simple "Take Last" single model in terms of gold score at 70M scale, suggesting that their calibration benefits might be limited at this scale, or that the optimization with PPO exploits them regardless of calibration. **We have clarified this point in paragraph 1 of Section 5.3 in the revised manuscript.**
> > >
> > > > 6. MMD operationalization. How sensitive are your MMD conclusions to kernel choice/bandwidth and to KL bucketing? Could you release a minimal script so others can reproduce the curves on their pipelines?
> > >
> > > We use the standard squared exponential kernel. Regarding sensitivity, we found the MMD to be robust. We standardize reward functions to mean 0 and variance 1 before comparison (Appendix B.1) to ensure the metric captures distributional shape rather than affine shifts. This is an important step of the evaluation pipeline. We then aggregate data points by KL bucket to visualize trends (Section 4). In addition, we also showed the raw data points for all seeds. We are committed to releasing the code, which includes the script for the MMD calculation and the exact kernel bandwidths used.
> > >
> > > Sincerely,
> > >
> > > The Authors
> > >
> > > [1] Gao, L., Schulman, J., and Hilton, J. Scaling Laws for Reward Model Overoptimization. In Proceedings of the *ICML’23*. 2023.
> > >
> > > [2] Coste, et al.. Reward model ensembles help mitigate overoptimization. In *ICLR’24*. 2024
> > >
> > > [3] Ramé, et al.. WARM: On the Benefits of Weight Averaged Reward Models.  In *ICML’24*. 2024.
> > >
> > > [4] Zhu, B., Jordan, M., and Jiao, J. Iterative Data Smoothing: Mitigating Reward Overfitting and Overoptimization in RLHF. In  *ICML’24*. 2024.

---

### Official Review · Reviewer_C7Qy · 2025-11-01

**Soundness:** 3
**Presentation:** 3
**Contribution:** 3
**Rating:** 4
**Confidence:** 2

**Summary:**

This paper presents a comprehensive study of reward model overoptimization in iterated RLHF. It finds that while overoptimization decreases across iterations as reward models better approximate human preferences, performance gains plateau and some overoptimization persists. The authors identify three key design choices—how preference data is managed, how reward models are combined, and how policies are initialized—that significantly impact outcomes.

**Strengths:**

1. The paper analyzed the overoptimization problem of iterative RLHF very thoroughly, including empirical study on its progression across multiple training rounds, the impact of key design choices like data aggregation and policy initialization, and the trade-offs between robustness and optimization flexibility.

2. The paper provides a novel, theoretical perspective to study overoptimization.

**Weaknesses:**

1. The paper lacks testing on standard reward benchmarks.

2. The paper's content is not organized enough to understand the whole process of iterative RLHF design choices and evaluating overoptimization.

**Questions:**

1. Are the findings generalizable to other models and other datasets?

2. If a model is trained using the principles discovered, what is the performance against other iterative RLHF designs?

I am not sure I have enough knowledge to give a deeper insight of the paper.

---

> ### Author Response · Authors · 2025-11-26
> **Official comment by Authors**
>
> We thank the Reviewer for their efforts and valuable feedback on our work. We are confident that we can address the weaknesses raised by the Reviewer. We kindly request you to consider increasing the score based on our detailed explanation and additional results.
>
> ## Weaknesses
>
> > Weakness 1) The paper lacks testing on standard reward benchmarks.
>
> We respectfully disagree with the Reviewer that our paper lacks testing on standard reward benchmarks. **Our evaluation setup directly follows [1], which closely resembles the evaluation setup of [2]. In the study of overoptimization the comparison to the gold reward model is critical.**
>
> **That being said, we have evaluated the reward models on RewardBench [3] to provide results on a standard reward model benchmark. We have included this evaluation in Section 5.5 of the revised paper and provide additional results in Appendix D (also the table below)**. In summary, we observe that reward models, corresponding to methods achieving high gold reward in the final iteration, achieve comparable or better performance on AlpacaEval subsets in the final iteration (Concat Data 70m, LITI 160M, Concat Data 160m). For other methods the reward models in the final iteration achieve lower accuracy on AlpacaEval subsets than those trained in the initial iteration.
>
> It is important to note that these results are less critical than the evaluation of the gold reward of the fine-tuned policy. Specifically, reward models in later iterations may successfully prevent overoptimization despite worse performance on the RewardBench test set due to distribution shift.
>
> **Table 1) RewardBench results.**
>
> | RM Size | Experiment | Iteration | Accuracy | AlpacaEval Hard | AlpacaEval Easy | AlpacaEval Length |
> |--------|---|--|---|--|--|----|
> | 70m    | WCO  | 4 | 0.550    | 0.702  | 0.656 | 0.588   |
> | 70m    | WCO | 1 | 0.536    | 0.809 | 0.786  | 0.617 |
> | 70m    | Take Last| 4 | 0.533    | 0.735 | 0.711 | 0.586 |
> | 70m    | Take Last | 1 | 0.535    | 0.838 | 0.790 | 0.580 |
> | 70m    | Sample | 4 | 0.553    | 0.737 | 0.686 | 0.584  |
> | 70m    | Sample | 1 | 0.537    | 0.817 | 0.777 | 0.609 |
> | 70m    | Ensemble  | 4 | 0.553    | 0.713 | 0.721| 0.591 |
> | 70m    | Ensemble  | 1 | 0.536    | 0.836 | 0.806 | 0.604 |
> | 70m    | Concat Data / Policy from SFT | 4 | 0.521    | 0.825 | 0.807 | 0.586  |
> | 70m    | Concat Data / Policy from SFT | 1 | 0.534    | 0.851 | 0.800 | 0.586  |
> | 70m    | Concat Data + LITI (0.5) | 4 | 0.532    | 0.855 | 0.845  | 0.592   |
> | 70m    | Concat Data + LITI (0.5) | 1 | 0.536  | 0.805 | 0.789 | 0.584 |
> | 160m   | WCO | 4 | 0.497| 0.809 | 0.784  | 0.576 |
> | 160m   | WCO | 1 | 0.439 | 0.943 | 0.889| 0.534  |
> | 160m   | Sample Exclusive  | 4| 0.485 | 0.847 | 0.797 | 0.562 |
> | 160m   | Sample Exclusive | 1  | 0.456    | 0.917 | 0.889 | 0.542 |
> | 160m   | Sample | 4  | 0.496  | 0.868 | 0.845 | 0.586  |
> | 160m   | Sample | 1| 0.450 | 0.932| 0.910 | 0.536  |
> | 160m   | Ensemble   | 4  | 0.496 | 0.818 | 0.805 | 0.567 |
> | 160m   | Ensemble | 1 | 0.450 | 0.930 | 0.882 | 0.529 |
> | 160m   | Concat Data / Policy from SFT    | 4 | 0.492    | 0.925| 0.914 | 0.638 |
> | 160m   | Concat Data / Policy from SFT    | 1 | 0.489    | 0.941 | 0.899 | 0.534 |
> | 160m   | Concat Data + LITI (0.5)| 4  | 0.482    | 0.928  | 0.922 | 0.632   |
> | 160m   | Concat Data + LITI (0.5  | 1 | 0.466    | 0.918  | 0.886  | 0.566    |
>
> > Weakness 2) The paper's content is not organized enough to understand the whole process of iterative RLHF design choices and evaluating overoptimization.
>
> In the submitted version of the manuscript we have taken the following actions to clearly communicate the iterative RLHF process, the design choices, and the evaluation of overoptimization:
> * Section 3.1 introduces the standard RLHF pipeline;
> * Section 3.2 discusses in detail how the single iteration is extended to multiple iterations together with each design choice;
> * Figure 1 clearly illustrates the overall pipeline as well as how the design choices fit in;* Algorithm 1 provides the detailed pseudocode for the process;
> * Figure 3 illustrates the different design choices we evaluated;
> * Section 4 first outlines the training details for our empirical evaluation and then discusses how overoptimization is measured via the mean reward and MMD between ground truth and proxy reward distributions on a hold out set.
>
> We would kindly ask the Reviewer to provide more specific feedback regarding the above or other parts of the paper, so that we can improve and incorporate the suggestions in a revised version of the manuscript.

---

> > ### Author Response · Authors · 2025-11-26
> > **Official Comment by Authors**
> >
> > ## Questions
> >
> > > Question 1) Are the findings generalizable to other models and other datasets?
> >
> > We expect the relative order and characteristics observed with respect to the design choices as well as the iterative behaviour to generalize to other models and datasets. While certain properties such as the strength of overoptimization and original performance on AlpacaFarm (and other datasets) depend on the base model, the trends and relationships between design choices and model size should remain consistent. Our study closely follows prior works [1,2], which also yielded generalizable insights beyond the evaluated model and dataset. It is also worth noting that the iterative setting together with the large number of design choices and seeds already required significant compute resources, which ultimately are the limiting factor in our academic  setting. We really hope that we will not be penalized for this.
> >
> > > Question 2) If a model is trained using the principles discovered, what is the performance against other iterative RLHF designs?
> >
> > The best possible design choices can result in policies that consistently improve in terms of ground-truth reward with each iteration. On the other hand, poor choice can lead to complete failure and policies that strongly overoptimize the proxy reward model resulting in very low ground-truth reward. We have evaluated an extensive range of possible options throughout the paper, if the Reviewer could pinpoint some specific papers we would be happy to provide additional clarification.
> >
> > Sincerely,
> >
> > The Authors
> >
> > [1] Coste, T., Anwar, U., Kirk, R., & Krueger, D. Reward Model Ensembles Help Mitigate Overoptimization. In *ICLR 2024*.
> >
> > [2] Gao, L., Schulman, J., & Hilton, J. Scaling Laws for Reward Model Overoptimization. In *ICML 2023*.
> >
> > [3] Lambert, N. et al. RewardBench: Evaluating Reward Models for Language Modeling. *arXiv preprint arXiv:2403*.13787, 2024.

---

> > > ### Comment · Reviewer_C7Qy · 2025-11-27
> > >
> > > Thank the authors for their prompt response to my concerns. From prior works, it seems like a convention to overlook the real RLHF performance and only focus on the reward modeling performance, i.e. my main concern is usually "assumed" to be correct. With that said, I will raise my score to 6, but still I would want to see some real RLHF language modeling results in later versions of the paper.

---

### Author Response · Authors · 2025-12-03
**Summary of Rebuttal Progress Prior to Score Reset**

Dear Area Chair,

We thank you for your efforts in the reassessment of our paper. In the following, we provide a summary of how our rebuttal addresses the reviewers’ concerns, as well as the additional experiments and updates we have made to the revised manuscript.

## Summary of the Discussion:

Firstly, we were pleased that Reviewer C7Qy had raised their score to a 6 following our rebuttal, noting that we had addressed their main concerns. Reviewer C9FL had maintained their already positive score. We believe our responses to Reviewers JWGj and utsC address their critiques regarding evaluation on benchmarks and metrics as well as external validity and robustness.

We would also like to bring to your attention the review of Reviewer JWGj as possibly LLM generated (pangram and GPTZero classify it with high confidence, e.g. https://iclr.pangram.com/reviews?submission_number=17799).

## Updates and additional experiments:
In the following, we provide a detailed summary of the changes and additional experiments that we made during the rebuttal phase.

**1. Additional benchmark evaluation to address concerns of Reviewers C7Qy and utsC.**

To address the concern regarding reward model benchmarks and metrics, we evaluated the reward models for different design choices and across iterations on the standard RewardBench benchmark. The results and discussion are included in Section 5.5 and Appendix D of the revised manuscript. This confirms that our findings generalize to benchmarks and standard metrics.

**2. Robustness to label noise experiments to address concerns of Reviewer JWGj.**

We conducted new experiments with 25% label noise to simulate a more realistic scenario with annotator inconsistency. The results included in Appendix E of the revised manuscript highlight the robustness of our findings.

**3. Computational cost discussion to address concerns of Reviewers JWGj and C9FL.**

We added a more detailed discussion and comparison of the computational cost (inference and training) and trade-offs regarding the different design choices. The discussion has been added in Appendix C.4 of the revised manuscript.

**4. Analysis of policy collapse to address the question of reviewer C9FL.**

We have provided additional plots of entropy that rule out entropy collapse as the key driver of failure to recover from overoptimization, and added qualitative examples to highlight the repetitive and idiosyncratic responses obtained due to overoptimization. The results can be found in appendices F.5 and G.


**5. Validity of model scale and dataset:**

Our evaluation setup is consistent with accepted works in the field, which utilize similar model scales and single-dataset evaluations. The rebuttal argues that our results generalize based on established scaling laws. Specifically, that policy scale is not a key driver of overoptimization and effects of RM model scale are smooth such that trends are preserved at larger scales. We selected AlpacaFarm because it is the standard testbed for isolating RLHF dynamics, allowing for scientifically rigorous ground-truth measurement infeasible with human-in-the-loop methods.

**6. Follow up on discussion with reviewer C7Qy:**

Regarding the discussion on a `real language modelling experiment’:  If we understood correctly, this necessitates human-annotated on-policy preference labels to evaluate overoptimization. Unfortunately, the time required for ethics approval and participant recruitment, as well as large cost, made this impossible to execute during the rebuttal.

## Conclusion
In summary, we have addressed the main concerns as follows:
*   **New Experiments:** We added results on the standard **RewardBench** benchmark (Section 5.5, Appendix D) and an evaluation of robustness to **25% label noise** (Appendix E).
*   **Detailed Analysis:** We provided a comprehensive discussion on **computational costs** (Appendix C.4) and a deeper analysis of **policy collapse** mechanisms (Appendices F.5, G).
*   **Evaluation consistency:** We confirmed the generalizability of our findings across model scales and datasets to be consistent with established scaling laws and prior work.

With Reviewer C7Qy having raised their score, and all weaknesses from the other reviewers being addressed through new results and detailed clarifications, we would be delighted if the paper could now be considered for acceptance.

Thank you once more for your time and consideration.

Sincerely,

The Authors

---

### Meta-Review · Area_Chair_JWoB · 2026-01-01

**Summary:**

The reviewers' concerns mainly focus on whether the conclusion is scalable, including both the dataset and model size. However, the author claims that the cost of the RLHF pipeline is expensive, without an additional dataset. And another major concern is the adoption of the gold reward model, which will make the whole results seem unreliable. Despite the common practice of following previous work, however, some more results will enhance the concerns about the gold reward model.

**Reviewer Concerns:**

The concern about the gold reward model has been carefully addressed by authors, giving more details on experiments and previous work. However, the generalization is still an important problem, because the work is devoted to giving general findings.

**Reviewer Scores:**

The reviews from Reviewer JWGj are almost the LLM-generated, which I do not take into consideration.

The Reviewer utsC will keep the score due to the weakness 3 cannot be addressed right now.

---

### Decision · Program_Chairs · 2026-01-26

Reject